# Control of telomerase recruitment and end protection by independent shelterin components

Ranjodh Sandhu[1], Gianna M. Tricola[1], Si Young Lee[1], Andy Tran [2] & Eros Lazzerini Denchi [1]✉

Telomeres are proposed to alternate between "closed" states, in which chromosome ends are protected from DNA damage signaling and inaccessible to telomerase, and "open" states, where they become accessible for telomerase mediated elongation but less protected. Whether these states reflect distinct molecular mechanisms or mutually exclusive structural conformations remains unclear. Here, we develop a single-cell assay to monitor telomerase activity in mouse embryonic stem cells. Using this approach, we demonstrate that the shelterin component TPP1 is essential for telomerase recruitment via its interaction with TIN2, independently of POT1. In contrast, POT1 is dispensable for telomerase function but required for telomere end protection, acting independently of TPP1. These findings challenge the classical open-closed telomere model and reveal that telomerase recruitment and end protection are mediated by genetically and molecularly separable mechanisms.

Mammalian telomeres are specialized nucleoprotein structures that cap chromosome ends and prevent inappropriate activation of the DNA damage response pathway. Telomere function is ensured by the shelterin complex, a six-subunit assembly that binds double- and single-stranded telomeric DNA. A central question in the telomere biology field is how shelterin simultaneously permits periodic telomerase access while maintaining continuous protection against aberrant DNA damage activation.

A proposed solution for this dilemma is an "open-closed" model of telomeres, in which chromosome ends transition between distinct structural states[1–3]. In the "closed" configuration, telomeres are protected from DNA damage signaling and refractory to telomerase, while in the "open" state, telomerase can engage the overhang, but at the cost of transient loss of end protection. This model is supported by observations in both yeast and mammalian systems, where telomerase accessibility has been associated with disruption of telomere protection and activation of a transient DNA damage response[4–7]. However, it remains unclear whether these states reflect mechanistically distinct pathways, or a dynamic conformational change that balances elonga-

tion and protection. A major limitation to address this fundamental question has been the resolution limits of available assays to detect telomerase activity.

TPP1-POT1 heterodimer within the shelterin complex is uniquely positioned to coordinate both protection and telomerase accessibility. TPP1 recruits telomerase via its TEL patch residues while POT1 binds the single-stranded G-overhang and plays an essential role in suppressing ATR-mediated DNA damage signaling[8–16]. Together, TPP1 and POT1 also recruit CST complex to telomeres, which mediates the termination of telomerase activity and promotes fill-in synthesis of the C-strand[17–20]. By integrating telomerase recruitment with end protection, the TPP1-POT1 heterodimer has been proposed to function as a molecular switch that governs the dynamic open or closed states of telomeres[8]. However, direct in vivo evidence supporting this model remains limited, largely due to the interdependent nature of TPP1 and POT1 activities. TPP1-POT1 complex enhance telomerase activity in vitro[8], and recent structural studies reveal that each can interact directly with telomerase[21]. Similarly, both are essential for telomere end protection[22–24]. These overlapping functions have made it chal-

[1]Laboratory of Genome Integrity, National Cancer Institute, NIH, Bethesda, MD, USA. [2]Confocal Microscopy Core Facility, National Cancer Institute, NIH, Bethesda, MD, USA. ✉e-mail: eros.lazzerinidenchi@nih.gov

lenging to understand their individual contributions, and the mechanistic link between telomerase regulation and end protection remains unresolved.

Here, we address this question by developing a novel single-cell telomerase activity assay to independently measure telomerase recruitment and end protection in vivo using mouse embryonic stem cells (mESCs). Using this system, we dissect the roles of POT1 and TPP1 and show, for the first time, that recruitment of telomerase activity and telomere protection are mechanistically separable functions of the TPP1-POT1 dimer. Our findings challenge the open/closed binary model and instead reveal a modular architecture in which elongation and protection are independently regulated by distinct shelterin components.

## Results

### iTAP: a telomerase activity assay with single-cell resolution
A fundamental challenge in dissecting the relationship between telomerase recruitment and end protection has been the lack of a system to monitor telomerase activity at single-cell resolution. To address this, we developed a live-cell assay in which telomerase-mediated elongation can be tracked through the incorporation of mutant telomeric repeats. We refer to this system as iTAP (inducible Telomerase Activity Probing). We engineered a doxycycline (DOX)-inducible system to express a mutant telomerase RNA (mutTR) carrying the 47/53 A mutation, which serves as an altered template for telomerase, leading to the incorporation of mutant telomeric repeats (TTTGGG) that can be distinguished from the canonical TTAGGG sequence[25]. We reasoned that incorporation of these mutant repeats would provide a direct and quantitative readout of telomerase activity at chromosome ends.

We implemented this system in mESCs, which exhibit robust endogenous telomerase activity[26]. Stable clonal lines carrying the inducible mutTR transgene showed a 10-15-fold induction of expression upon DOX treatment, with minimal background in uninduced cells (Supplementary Fig. 1A). Importantly, mutTR induction did not impair cell proliferation (Fig. 1A) or activate a telomeric DNA damage response (Supplementary Fig. 1B). In other cell types constitutive expression of mutTR at higher levels than endogenous hTR has been shown to induce a strong DNA damage response at telomeres[27–29]. We speculate that, in our system, mutTR expression can be tolerated since it is coinfined to a short window of time and expressed at relatively low levels compared to endogenous TR. In support of this hypothesis induction of mutTR in cells depleted for endogenous TERC was highly toxic, resulting in growth arrest (Supplementary Fig. 1C).

To determine whether this approach could be used to detect telomerase activity, we performed fluorescence in situ hybridization (FISH) on interphase cells using probes specific for either wild-type telomeric repeats (TTAGGG) or mutant repeats (TTTGGG). After 64 h of DOX induction, the majority (~80%) of cells showed robust incorporation of mutant telomeric repeats, in stark contrast to uninduced controls (Fig. 1B, Supplementary Fig. 1D, E). Importantly, incorporation was completely abolished in cells lacking TERT, confirming the specificity of the assay (Fig. 1B). As expected, mutant repeat incorporation occurred at chromosome ends, as shown by FISH on metaphase spreads (Fig. 1C).

To further validate the assay, we employed two orthogonal approaches: (i) telomere restriction fragment (TRF) analysis following mutTR induction (Supplementary Fig. 1C), and (ii) DNA dot blot hybridization using a mutant repeat-specific probe (Fig. 1D). Both approaches confirmed TERT-dependent incorporation of mutant repeats (Fig. 1C, D).

Finally, to directly assess the synthesis of mutant telomeric repeats, we performed next-generation sequencing (NGS). To enrich telomeric DNA, we applied a strategy based on digestion with frequent-cutting restriction enzymes, which selectively removes most of the genomic DNA. This approach, previously used for telomeric DNA isolation for electron microscopy[30], is here adapted for NGS and referred to as Frequent Cutter Resistance sequencing (FCR-seq). FCR-seq enabled approximately a 30-fold enrichment of telomeric reads compared to their expected representation in the mouse genome (~25,000 reads per million observed versus the expected 741 reads per million, Supplementary Fig. 1D). Quantification of mutant (TTTGGG) versus wild-type (TTAGGG) telomeric repeats shows a significant incorporation of the mutant repeats in cells expressing the mutTR (Fig. 1E). Collectively, these results establish iTAP as a quantitative assay to detect telomerase activity.

### TPP1, but not POT1, is required for telomerase activity in vivo
Having established a single-cell assay to measure telomerase activity without compromising cell viability or telomere integrity, we applied this method to dissect the contribution of the TPP1-POT1 heterodimer to telomerase function. TPP1 and POT1 form a complex that binds single-stranded telomeric DNA, with TPP1 interacting directly with telomerase via its TEL patch. While POT1 has been proposed to regulate telomerase processivity and access, its in vivo role remains unclear.

CRISPR-Cas9-mediated depletion of *Tpp1* (Supplementary Fig. 2A, B) completely abolished the incorporation of mutant telomeric repeats, as assessed by interphase FISH (Fig. 2A), DNA dot blot hybridization (Fig. 2C), FCR-seq (Fig. 2E), and telomere restriction fragment (TRF) analysis (Supplementary Fig. 2J). Notably, TPP1-depleted cells show mutTR expression comparable to those in TPP1-proficient cells upon DOX addition (Supplementary Fig. 2I).

To assess whether POT1 contributes to telomerase activity, we depleted both *Pot1* paralogs, *Pot1a* and *Pot1b*, which are co-expressed in mouse cells. We generated individual knockout lines for *Pot1a* and *Pot1b* (Supplementary Fig. 2C, D), as well as a degron-based system to achieve inducible degradation of *Pot1a* in the presence or absence of *Pot1b* (Supplementary Fig. 2F–H[31]). For this, we tagged the endogenous *Pot1a* gene with a FLAG-FKBP12$^{F36V}$ degron cassette, allowing both detection and inducible degradation. Treatment with the small molecule dTAG-13 efficiently depleted POT1a within 5 h (Supplementary Fig. 2H).

Using this system, we evaluated the contribution of each paralog individually and in combination to telomerase activity. Strikingly, neither deletion of *Pot1a* or *Pot1b*, nor loss of both, had a measurable impact on the incorporation of mutant telomeric repeats, as assessed by interphase FISH (Fig. 2B; Supplementary Fig. 2E), DNA dot blot (Fig. 2D; Supplementary Fig. 3B), FCR-seq (Fig. 2E), and TRF analysis (Supplementary Fig. 3A). These findings indicate that POT1 is dispensable for telomerase recruitment and activity in vivo. In contrast, loss of either individual paralog-or both-resulted in a 25–30% increase in mutant repeat incorporation by FCR-seq (Fig. 2E), suggesting that POT1 acts to limit telomerase activity in vivo, in agreement with previous reports[25,32,33]. To determine whether this increase reflects enhanced telomerase processivity or increased access to the telomere, we analyzed the number of mutant repeats added in *Pot1a*-proficient and *Pot1a*-deficient cells using FCR-seq. Our analysis revealed that in the absence of *Pot1a*, there is a general increase in the number of mutant repeats across all repeat lengths without significant increase in longer products. (Supplementary Fig. 3C). These findings suggest that *Pot1a* loss leads to increased telomerase engagement at telomeres, rather than enhanced processivity of the enzyme. Of note, POT1 cannot bind efficiently to the mutant telomeric sequences[34], suggesting that our assay may underestimate the effect of POT1 on telomerase once this enzyme is engaged with the substrate. Nevertheless, in our inducible experimental setting we are primarily assessing the first round of incorporation of mutant repeats within a wild-type telomeric sequence, thus limiting as much as possible this caveat.

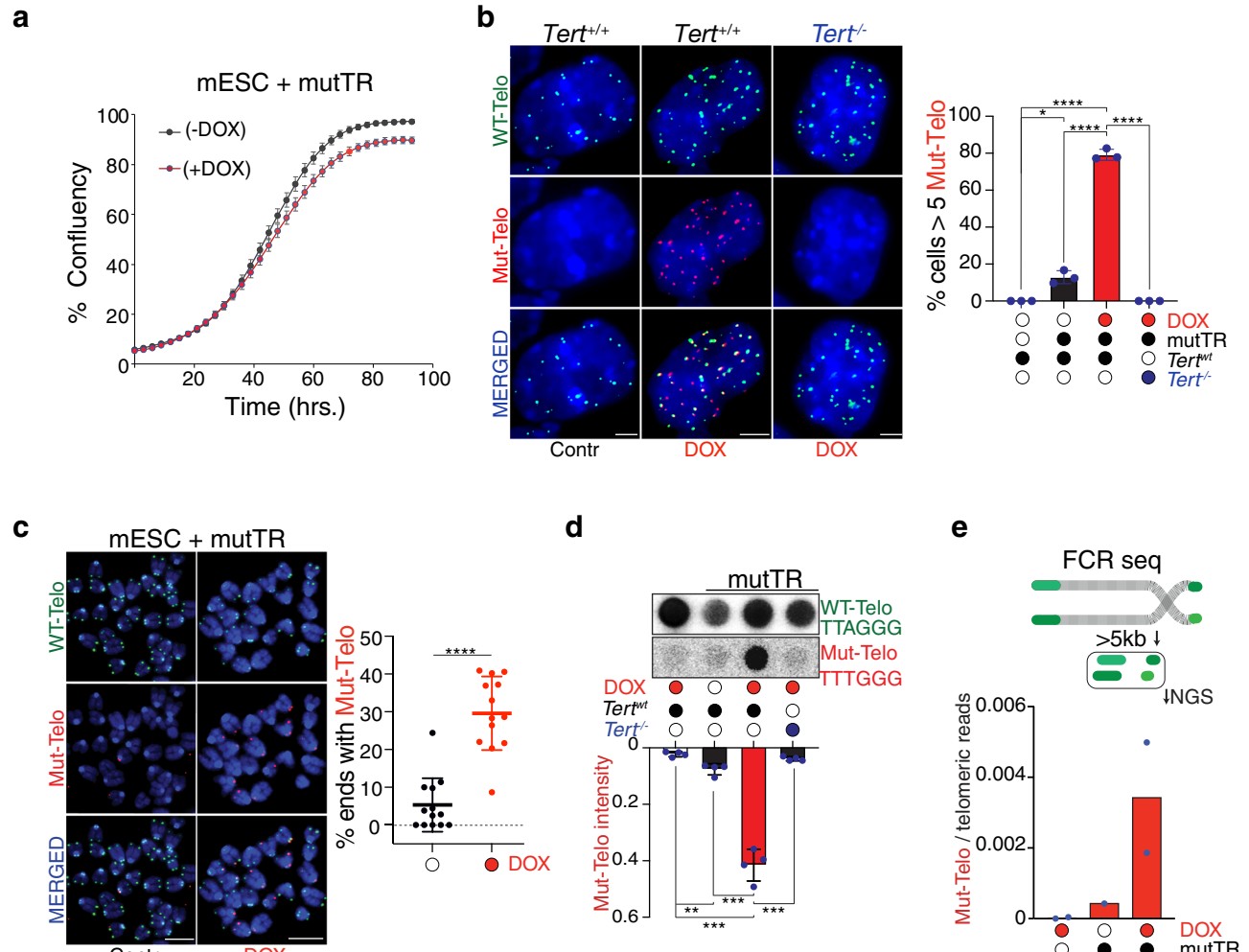

**Fig. 1 | A telomerase activity reporter reveals mutant repeat incorporation in vivo. a** Proliferation of mESCs carrying an inducible mutant telomerase RNA (mutTR) transgene was monitored following doxycycline induction (+DOX) or under control conditions (−DOX) by measuring confluence over time using the IncuCyte S3 system. Data show mean ± s.d. from three independent biological replicates. **b** PNA-FISH analysis of mESCs of the indicated genotypes using probes specific for wild-type telomeric repeats (WT-Telo, [TTAGGG], green) and mutant repeats (Mut-Telo, [TTTGGG], red). Quantification shows the percentage of cells positive for mutant telomeric foci after 64 h of mutTR induction, defined as ≥5 Mut-Telo foci per cell. Bars represent mean ± s.d. from n = 3 biological replicates. Two-tailed unpaired t test (* indicates P = 0.0239; **** indicates P < 0.0001).
**c** Representative metaphase spreads from uninduced (−DOX) and induced (+DOX) mESCs expressing mutTR, showing FISH detection of wild-type and mutant telomeric repeats. Scatter plot shows the percentage of telomeres positive for mutant repeats per metaphase after 64 h of induction. Each dot represents one metaphase. Data are mean ± s.d. from three biological replicates; two-tailed unpaired t test (**** P < 0.0001). **d** Dot blot analysis of genomic DNA from wild-type or TERT-deficient mESCs harboring inducible mutTR, untreated or treated with doxycycline for 64 h, hybridized with probes specific for wild-type or mutant telomeric repeats. Quantification shows the ratio of mutant to wild-type telomeric signal intensity. Data represent mean ± s.d. from three biological replicates; two-tailed unpaired t test (** P = 0.0025, *** P = 0.0005). **e** Schematic of the FCR-seq approach. Telomeric DNA is shown in green, non-telomeric DNA in grey, and gaps denote frequent-cutter restriction sites. Graphs show the fraction of sequencing reads containing mutant telomeric repeats relative to total reads with ≥5 telomeric repeats after 64 h of mutTR induction (two biological replicates). All microscopy images include 5 µm scale bars. Mean values and numbers of nuclei analyzed per replicate are provided in the Source data file.

## TPP1-mediated telomerase recruitment requires interaction with TIN2 but not POT1

TPP1 is known to interact with: (i) telomerase through its OB-fold domain, (ii) POT1 through its RD domain, and (iii) TIN2 through its C-terminal region[10,35–37]. To determine which of these interactions is required for telomerase activity, we performed complementation assays in TPP1-deficient mESCs using the iTAP assay.

*Tpp1*-deficient cells were complemented with either full-length *Tpp1* (WT), a truncated construct encoding only the OB domain (OB), or mutant variants defective in either POT1 binding (TPP1^ΔRD) or TIN2 binding (TPP1^ΔTIN2) (Fig. 2F−H). Expression of full-length TPP1 restored telomerase activity to levels comparable to those observed in *Tpp1*-proficient cells (Fig. 2I). In contrast, expression of the OB construct

failed to rescue telomerase activity (Fig. 2I), indicating that the OB domain alone, and thus the TPP1-TERT interaction, is not sufficient to promote telomerase function at telomeres as reported earlier[10,35].

Expression of TPP1^ΔRD, which localizes to telomeres but cannot bind POT1, fully restored telomerase activity to the same extent as wild-type TPP1 (Fig. 2I), demonstrating that the TPP1-POT1 interaction is dispensable for telomerase function, in agreement with the POT1-deletion data described above (Fig. 2B). By contrast, the TPP1^ΔTIN2 variant was unable to rescue telomerase activity, suggesting that TPP1's ability to stimulate telomerase requires its recruitment to telomeres via interaction with the shelterin component TIN2.

Collectively, these data demonstrate that TPP1-mediated telomerase recruitment and activation in vivo requires its interaction with

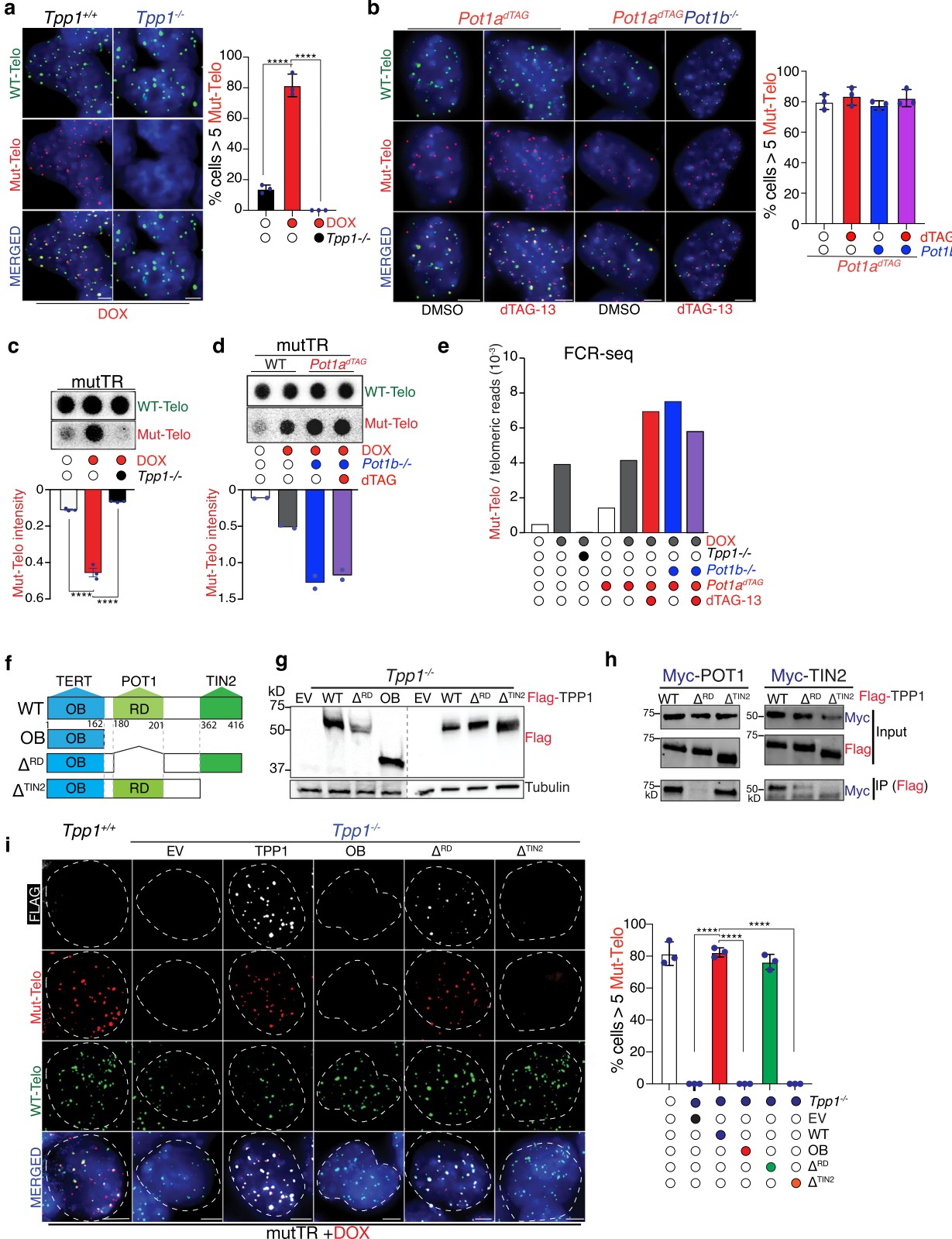

TIN2, but not with POT1. This data further supports the notion that the telomerase recruitment function of TPP1 is independent and distinct from its role in interaction with POT1.

## POT1 is essential for end protection, TPP1 is not

Given the distinct roles of TPP1 and POT1 in telomerase recruitment and activity, we next asked whether the TPP1-POT1 complex is required for telomere end protection. Strikingly, while *Tpp1*-deficient cells are viable and show no significant proliferation defect (Fig. 3A), co-depletion of *Pot1a* and *Pot1b* led to a dramatic reduction in cell proliferation (Fig. 3B). This finding is further supported by CRISPR-Cas9 gene editing results, where recovery of homozygous deletions was readily achieved for *Tpp1* but not for *Pot1*, suggesting that POT1 is essential for cell viability (Supplementary Fig. 4A).

**Fig. 2 | TPP1 is essential for telomerase function; POT1 is dispensable. a** TPP1-proficient and TPP1-deficient mESCs were hybridized with PNA probes complementary to wild-type telomeric repeats (WT-Telo, [TTAGGG], green) and mutant telomeric repeats (Mut-Telo, [TTTGGG], red). The graph shows the fraction of cells positive for mutant telomeric foci, defined as cells containing ≥5 Mut-Telo foci. Data represent mean ± s.d.; statistical significance was assessed using a two-tailed unpaired *t* test (**** *P* < 0.0001). **b** POT1b-proficient or -deficient cells expressing POT1a–FKBP12$^{F36V}$ were treated with DMSO or dTAG-13 to induce POT1a degradation and stained as in (**a**). The graph shows the fraction of cells with ≥5 Mut-Telo foci (mean ± s.d., three biological replicates). **c** Genomic DNA from wild-type or TPP1-deficient cells, untreated or treated with doxycycline (DOX), was hybridized with probes recognizing WT or mutant telomeric repeats. Graphs show the ratio of mutant-to-wild-type telomeric signal intensity. Data represent mean ± s.d.; statistical significance was assessed using a two-tailed unpaired *t* test **** *P* < 0.0001) from three independent experiments. **d** Genomic DNA isolated from wild-type or *Pot1a*$^{dTAG}$;*Pot1b*$^{-/-}$ cells, untreated, DOX-treated or dTAG-13 (dTAG)-treated was analyzed as in (**c**). Data represent mean of two independent experiments. **e** Fraction of telomeric reads containing mutant repeats following 64 h of mutTR induction, measured by FCR sequencing. Genotype and treatments are indicated. **f** Schematic of TPP1 domains, protein interactions, and alleles used in this study. **g** Immunoblot analysis of whole-cell lysates from mESCs expressing FLAG-tagged TPP1 alleles. **h** Co-immunoprecipitation of FLAG-tagged TPP1 alleles with MYC-tagged POT1a in 293 T cells. FLAG immunoprecipitates were immunoblotted for MYC to detect POT1a. **i** Cells of the indicated genotypes expressing the specified constructs were stained for FLAG, mutant telomeric repeats (red), and wild-type telomeric repeats (green). The graph shows the fraction of cells with ≥5 Mut-Telo foci. Data represent the mean ± s.d. from three independent biological replicates. Statistical significance was assessed using a two-tailed unpaired *t*-test; * indicates *P* < 0.0001. All microscopy images include 5 μm scale bars. Mean values and numbers of nuclei analyzed per replicate are provided in the Source data file.

To determine whether this difference reflects a specific role in end protection, we assessed telomere-associated DNA damage by quantifying cells positive for 53BP1 foci colocalizing with telomeres—a marker of telomere deprotection referred to as telomere dysfunction-Induced Foci (TIFs). TPP1-deficient cells exhibited minimal TIFs, whereas cells depleted of POT1a or both POT1 paralogs accumulated 53BP1 at telomeres in the vast majority of cells (Fig. 3C). In line with previous studies, this DNA damage response in *Pot1*-depleted cells was accompanied by increased RPA accumulation at telomeres (Supplementary Fig. 4B) and was dependent on the ATR kinase (Fig. 3D)[38].

Moreover, *Pot1*-depleted cells exhibited high levels of telomere fusions (Fig. 3F, G), which were completely suppressed by ATR inhibition (Fig. 3G). In contrast, *Tpp1*-null cells did not display elevated levels of telomere fusions (Fig. 3E, G). These findings argue against an obligate function for the TPP1-POT1 heterodimer in end protection and instead support the conclusion that *Tpp1* and *Pot1* have genetically and mechanistically separable functions.

## POT1-mediated end protection is independent of TPP1 interaction

Our finding that *Tpp1* and *Pot1* have genetically, and mechanistically distinct functions presents a conundrum, as POT1 recruitment to telomeres is generally thought to depend on its interaction with TPP1[39,40]. To determine whether TPP1 is similarly required for POT1 localization in mESCs, as previously reported in immortalized MEFs[39], we analyzed the localization of FLAG-dTAG-tagged POT1a in *Tpp1*-proficient and *Tpp1*-deficient backgrounds. This analysis revealed that the bulk of POT1a recruitment to telomeres is indeed TPP1-dependent: in the absence of TPP1, POT1a is largely undetectable at telomeres (Fig. 4A). Similarly, telomere accumulation of POT1b is dependent on TPP1. Overexpression of MYC-tagged POT1a or POT1b fails to localize to telomeres in the absence of TPP1 in contrast to their robust telomeric colocalization to telomeres in TPP1 proficient cells (Supplementary Fig. 4C). These findings are consistent with prior work and support a model in which TPP1 facilitates the stable recruitment of POT1 to chromosome ends. *Tpp1*-deficient mESCs exhibit impaired differentiation potential and fail to differentiate into neural progenitor cells (NPCs) (Supplementary Fig. 4D). These findings suggest that while TPP1 is dispensable for telomere protection in mESCs, it is required for POT1-mediated telomere end protection in differentiated cells.

However, this lack of POT1 accumulation at telomere in absence of TPP1 does not exclude the possibility that low levels of POT1a or POT1b bind directly to the single-stranded telomeric overhang in the absence of TPP1. This is plausible given that both paralogs contain OB-fold domains capable of directly recognizing single-stranded telomeric DNA. We hypothesized that this direct DNA binding may be sufficient to recruit enough POT1 to ensure telomere end protection, even in the absence of TPP1. To directly test this, we used a complementation approach in *Pot1a*$^{dTAG}$*Pot1b*$^{-/-}$ mESCs. dTAG-mediated degradation of POT1a in these cells led to robust activation of the DNA damage response, as evidenced by colocalization of 53BP1 with the telomeric marker TRF1 (Fig. 4E). Re-expression of full-length POT1a efficiently suppressed this damage response, whereas expression of a *Pot1a* mutant lacking the OB-fold domain (POT1a$^{ΔOB}$) failed to do so (Fig. 4E). In contrast, expression of a TPP1-binding-deficient mutant of *Pot1a* that retains the OB-fold domain (POT1a$^{ΔTPP1}$)[41] failed to accumulate at telomeres (Supplementary Fig. 5A–C) but suppressed the DNA damage response to a similar extent as full-length POT1a (Fig. 4E). These results demonstrate that POT1a's ability to protect telomeres is dependent on its capacity to bind single-stranded DNA and does not require interaction with TPP1. Finally, we tested whether the unique ability of POT1a to bind the telomeric double stranded- and single stranded DNA junction was required for end-protection. POT1a has a unique set of residues within the OB1 domain termed "POT-hole"[16]. Expression of a mutant allele of *Pot1a* lacking the POT-hole or *Pot1b* reduced TIF formation relative to the empty vector but failed to suppress it to wild-type POT1a levels (Fig. 4E Supplementary Fig. 5A, B). The resulting telomere damage was comparable to that observed upon *Pot1a* depletion (Fig. 3B), indicating that junction-binding activity is critical for full suppression of the telomeric DNA damage response. Although both the POT1a-hole mutant and POT1b were unable to fully protect telomeres, they rescued the growth defects of *Pot1a*/b-deficient cells (Fig. 4F), suggesting that this level of telomere damage is tolerated in mESCs. Finally, we tested whether expression of the OB fold domains of *Pot1a* were sufficient to suppress the DNA damage response at telomeres. Results of this experiment show that expression OB1 + OB2 of *Pot1a* (POT1a$^{OB1+OB2}$) is unable to prevent telomeric DNA damage (Fig. 4E Supplementary Fig. 5A, B), implicating the *Pot1a* C-terminus (POT1C) as a crucial contributor to telomere end protection beyond its interaction with TPP1.

Together, these findings functionally decouple the roles of TPP1 and POT1 and challenge the classical open-closed telomere model, which posits a reciprocal relationship between telomerase access and end protection. Instead, the distinct viability and DNA damage phenotypes observed in *Tpp1*- and *Pot1*-deficient cells support a modular model of telomere architecture, in which elongation and protection are governed by genetically separable mechanisms that do not operate in strict opposition.

## Discussion

Telomere maintenance depends on the ability to balance accessibility of telomerase to chromosome ends and persistent protection against DNA damage signaling. Current model integrates these states is the "open/closed" model, in which chromosome ends alternate between an "open" state permissive to telomerase and a "closed" state that enforces end protection. While conceptually compelling, direct in vivo

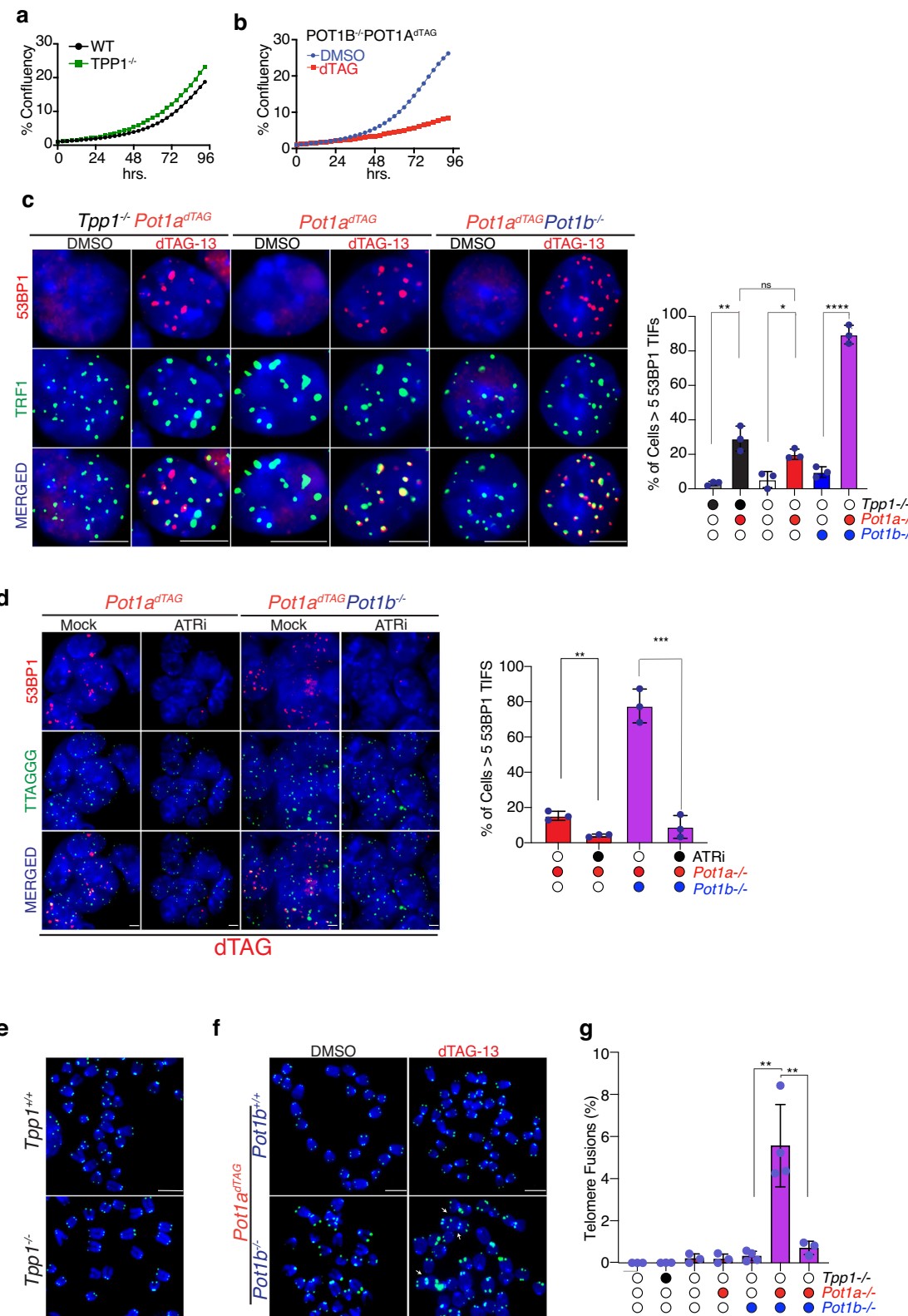

evidence for this dichotomy has been lacking, and it remains unclear whether telomerase accessibility and protection reflect distinct molecular mechanisms or mutually exclusive states.

Using a single-cell telomerase activity assay (iTAP), we addressed this dichotomy by dissecting the roles of two key shelterin components, TPP1 and POT1, which form a heterodimer at the single-stranded telomeric overhang. Our findings reveal that telomerase recruitment

and end protection in mouse embryonic stem cells are mediated by genetically separable and mechanistically distinct functions within the shelterin complex. We demonstrate that telomerase activity at telomeres strictly requires TPP1 but not POT1. Loss of *Tpp1* abrogates mutant repeat incorporation despite robust expression of mutant telomerase RNA, and this defect is rescued only by *Tpp1* variants capable of interacting with double stranded telomeric DNA through

**Fig. 3 | POT1, but not TPP1, is required for telomere end protection. a** Cell proliferation of TPP1-deficient and TPP1-proficient mESCs monitored by confluence using the IncuCyte S3 system. Data represent mean ± s.d. from three independent experiments. **b** Cell proliferation of cells expressing POT1a-FKBP12$^{F36V}$ and deficient for POT1b, treated with either DMSO or dTAG-13 to induce POT1a degradation. Proliferation was monitored as in (**a**). Data represent mean ± s.d. from three independent experiments. **c** Representative IF-FISH images showing 53BP1 (red) and telomeric DNA (green) in the indicated genotypes. Where indicated, dTAG-13 was used to induce degradation of POT1a-FKBP12$^{F36V}$. The graph shows quantification of cells with ≥5 telomere dysfunction-induced foci (TIFs). Data represent mean ± s.d. from three biological replicates. Statistical analysis was performed using two tailed unpaired $t$-test(* $P = 0.0103$; **= 0.0036**** $P < 0.0001$). **d** Representative IF-FISH images showing 53BP1 (red) and telomeric DNA (green) in POT1a$^{dTAG}$ cells with the indicated *Pot1b* genotype and treated with dTAG-13. Where indicated, cells were also treated with ATR inhibitor (ATRi) 1 μM AZ20 for 5 h. The graph shows quantification of cells with ≥5 TIFs. Data represent mean ± s.d. from three biological replicates. Statistical analysis by two-tailed unpaired $t$ test (** $P = 0.0020$; *** $P = 0.0005$. **e** Representative metaphase spreads showing FISH signals for wild-type telomeric repeats in control and TPP1-deficient mESCs. **f** Representative metaphase spreads showing telomeric FISH signals in POT1a$^{dTAG}$ cells with the indicated *Pot1b* genotype, treated with either DMSO or dTAG-13. Examples of telomere fusions are indicated by white arrows. **g** Quantification of chromosome ends engaged in telomere fusions for each genotype and treatment. Data represent mean ± s.d. from three independent experiments. Statistical analysis by two-tailed unpaired $t$ test (**$P < 0.01$. All microscopy images include 5 μm scale bars. Mean values and numbers of nuclei analyzed per replicate are provided in the Source data file.

TIN2 interaction. These results support a model in which TPP1 functions as a recruitment platform, connecting telomerase to the shelterin complex via TIN2 while TPP1-POT1 interaction is dispensable for this process (Supplementary Fig. 6). In agreement with this model, depletion of *Tin2*, but not *Pot1*, leads to defects in telomerase recruitment to telomeres[9] and in vitro tethering of TPP1 to telomere substrate via TIN2 can stimulate telomeres activity[42]. Conversely, loss of POT1 function results in telomere elongation due to increased telomerase activity, and mutations in POT1 that impair its DNA-binding domain similarly lead to a telomere overextension[32,43–45]. Depletion of *Pot1* does not prevent telomerase activity, strongly arguing against a positive role of POT1 in telomerase recruitment/activation or processivity. In contrast, although dispensable for telomerase function, POT1 plays a role in restricting telomerase activity. In the absence of *Pot1a* and *Pot1b*, telomerase activity is modestly increased, with an increase in incorporation events rather than an increase in processivity rates. We speculate that POT1 acts as a gatekeeper that limits telomerase engagement, though the single stranded telomeric DNA binding, a model that is in agreement with the telomere elongation phenotype of POT1 alleles lacking a DNA binding domain[7,33,46]. Lastly, the iTAP assay has the potential not only to address fundamental questions about telomerase regulation in cells, but also to serve as a tool for identifying novel modulators of telomerase activity

Although POT1 does not regulate telomerase recruitment, it is essential for telomere protection. Loss of both paralogs results in unprotected overhangs, RPA accumulation, ATR activation, and telomere fusions. Notably, this protective function does not require TPP1 binding, as POT1a mutants unable to interact with TPP1 still suppress DNA damage signaling when their OB-fold is intact. In agreement with this, while *Tpp1*-deficient cells remain viable, depletion of all *Pot1* paralogs results in loss of cell viability. Notably, this finding mirrors observations in human embryonic stem cells, where *Tpp1* depletion is tolerated but *Pot1* loss is not[47,48]. Our data demonstrate that, in mESCs, POT1-mediated end protection depends exclusively on its ability to bind single-stranded DNA, and not on its interaction with TPP1. Importantly, these results reveal a stark asymmetry in the consequences of losing *Tpp1* versus *Pot1*: *Tpp1*-null cells lack telomerase activity but remain viable and do not elicit a DNA damage response, whereas *Pot1*-deficient cells experience catastrophic telomere deprotection and cell death.

Together, our findings challenge the binary "open/closed" model and instead we propose a modular view of telomere regulation (Supplementary Fig. 6). In binary model, elongation and protection are thought to occur in mutually exclusive structural states. However, our data show that these functions can occur simultaneously and are governed by genetically separable activities within the TPP1-POT1 module, supporting the model for a modular system in which these functions are independently regulated. In this model, TPP1 and POT1 act in parallel but separable pathways: TPP1 bridges telomerase recruitment to telomeres via its interaction with TIN2, while POT1 binds directly the telomeric single stranded overhang to suppress ATR activation. The TPP1-POT1 heterodimer does not function as a molecular switch but instead integrates two distinct and independently controlled activities.

At the same time, our findings raise additional questions about how telomere length influences recruitment and protection. TPP1 is expected to be recruited efficiently at longer telomeres, due to the increased number of double-stranded binding sites for TRF1 and TRF2. As a result, longer telomeres may paradoxically have greater TPP1-dependent telomerase recruitment potential, despite not needing elongation. Furthermore, because POT1 is recruited in part through its interaction with TPP1, excess TPP1 at longer telomeres could sequester POT1 away from the overhang, potentially reducing its protective capacity. How this imbalance is resolved remains unclear and suggests that additional regulatory layers must exist to prioritize telomerase recruitment to short telomeres while preserving protection at long ones.

The uncoupling of telomere elongation and protection exposes a previously unappreciated complexity in how telomere length homeostasis is enforced. Rather than a simple feedback loop based on switching between structural states, it is likely that post-translational modification, subunit stoichiometry, or spatial compartmentalization further tune the activity of shelterin components across telomeres of different lengths.

## Method

### Cell culture
mESCs (E14) were grown on 0.1% gelatin-coated plates in DMEM medium containing LIF (1000 U/ml) and 2i (1 μM PD03259010 and 3 μM CHIR99021) as previously described[49]. When indicated cells were treated with 1 μg/mL doxycycline for 64 h and with 0.5 μM dTAG-13 (Millipore Sigma SML2601) for a minimum of 5 h.

### Differentiation of mouse ES cells in neural progenitor cells (NPC)
This procedure was adapted from refs. [50,51]. mESCs were cultured in ES cell medium supplemented with LIF and serum and dissociated using trypsin before plating onto gelatin-coated 10-cm dishes at $0.5 \times 10^6$ cells per dish. After 24 h, cultures were washed with PBS and switched to N2B27 medium lacking EGF and FGF for 7 days with daily medium changes, during which substantial cell death occurred. On day 7, cells were dissociated with TrypLE Express and $3 \times 10^6$ cells were transferred to 90-mm non–tissue-culture–treated dishes in N2B27 supplemented with EGF and FGF2 (10 ng/ml each) to generate free-floating neurospheres. On day 10, neurospheres were collected and plated onto gelatin-coated T75 flasks in N2B27 with EGF and FGF2, where attached spheres produced expanding NPC monolayers; flasks were gently tapped to remove non-adherent spheres once cultures approached confluence. NPCs were subsequently expanded for 2–3 passages, with filtration used as needed to remove aggregates. For routine maintenance, NPCs were cultured on 0.1% gelatin in N2B27

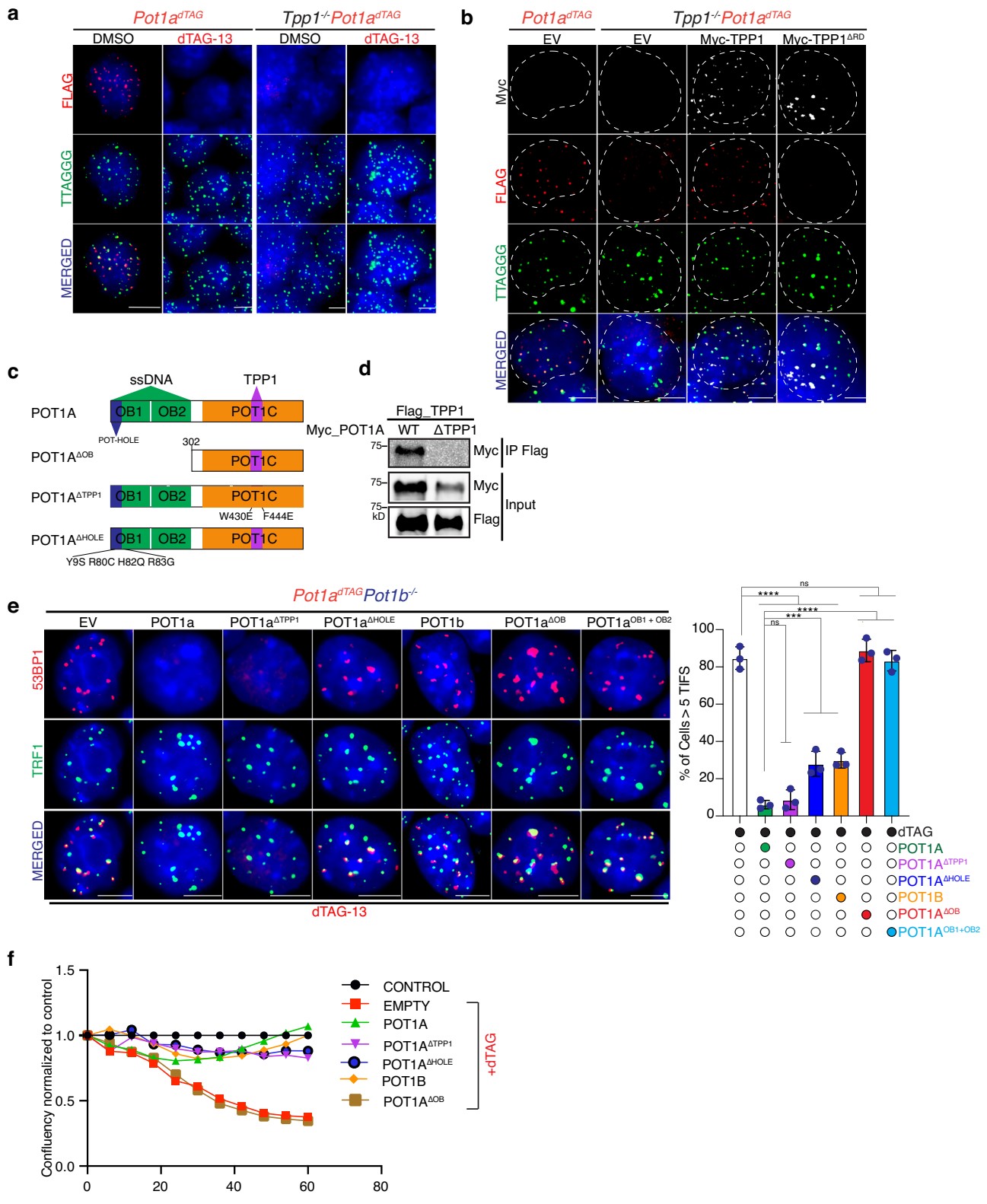

containing EGF and FGF2 (10 ng/ml each) and passaged every 2–3 days at ~80% confluence following 1–3 min Accutase dissociation, centrifugation at 1000 rpm for 3 min, and replating at a 1:2–1:4 ratio.

## Cell growth analysis
Cell proliferation was monitored in real-time using the IncuCyte S3 system (Essen Bioscience) based on confluence measurements.

mESCs were imaged every 3 h with a 10× objective. At least three biologically independent replicates were analyzed per condition.

## Inducible telomerase activity probing (iTAP)
A synthetic hTR construct containing 148 bp of the 3′ UTR and a mutation in the template region (47/53 A) was ordered as Geneblock (Bio Basic) and subcloned into the PB-TRE-dCas9-VPR plasmid

**Fig. 4 | ssDNA binding by POT1 is required for telomere protection in mESCs.**
**a** FISH-IF for FLAG (red) and FISH for telomeric DNA (green) in POT1a$^{dTAG}$ in TPP1 proficient or TPP1-deficient (Tpp1$^{-/-}$). Endogenous POT1a-FLAG-FKBP12$^{F36V}$ is shown in red (FLAG staining). Cells were treated with DMSO or with dTAG-13 to induce depletion of POT1a. **b** IF-FISH analysis of cells of the indicated genotypes ectopically expressing the indicated constructs. TPP1 allele expression is shown in white (MYC staining), endogenous POT1a-FLAG-FKBP12$^{F36V}$ is shown in red (FLAG staining), and telomeric DNA is shown in green. **c** Schematic representation of POT1a domains and interactions, along with the different POT1a alleles used in this study. **d** Co-immunoprecipitation of FLAG-tagged TPP1 with MYC-tagged POT1a alleles in 293 T cells. Lysates from cells co-expressing the indicated constructs were subjected to immunoprecipitation using a FLAG antibody, followed by immunoblotting with a MYC antibody to detect POT1a. **e** Representative IF images showing 53BP1 (red) and TRF1 (green) in POT1a$^{dTAG}$POT1b$^{-/-}$ cells and, where indicated, treated with dTAG-13. Cells were infected with either an empty vector (EV) or constructs expressing full-length POT1a, various POT1a mutants-including the TPP1-binding-deficient mutant (POT1a$^{ΔTPP1}$), the POT-hole mutant (POT1a$^{HOLE}$), the DNA-binding-deficient mutant (POT1a$^{ΔOB}$), and the OB domains of POT1a (POT1a$^{OB1+OB2}$) or full-length POT1b. The graph shows the quantification of cells with ≥5 TIFs. Data represent mean ± s.d. from three biological replicates. Statistical analysis was performed using two tailed unpaired *t*-test; *** indicates $P < 0.001$; **** indicates $P < 0.0001$. **f** Growth profile of POT1a$^{dTAG}$POT1b$^{-/-}$ cells infected with either an empty vector or the indicated POT1a or POT1b variants. Cells were treated with dTAG-13, and growth was normalized to DMSO-treated controls. All microscopy images include 5 µm scale bars. Mean values and numbers of nuclei analyzed per replicate are provided in the Source data file.

---

(Addgene #63800) to generate a PiggyBac-based, tetracycline-inducible expression vector. mESCs were co-transfected with this PiggyBac construct and a transposase-expressing plasmid using Lipofectamine 2000 for genomic integration. To minimize basal transgene expression, cells were cultured in tetracycline-free medium.

### CRISPR-Cas9 mediated genome editing
Two guide RNAs (gRNAs) targeting the genomic locus of the gene of interest were designed using the IDT online Guide Design Tool and cloned into the gRNA expression vector (Addgene #41824). Plasmids encoding gRNA spCas9 were co-transfected using Lipofectamine 2000 (Life Technologies) in mESCs and seeded a low density. The resulting colonies were screened for successful gene editing using locus-specific PCR and Sanger sequencing. To generate a *Pot1a* FKBP-tagged alleles we followed the same strategy with the addition of repair template containing homology arms, a blasticidin resistance gene, a 3xFLAG epitope, and an FKBP tag. Sequence for guide RNA's and primers used in screening can be found in source data.

### Immunofluorescence (IF) and IF-FISH
IF and IF-FISH were carried out as described in ref. 49 using the following primary antibodies: anti-TRF1 (Abcam 192629 1:1000), anti-FLAG (Sigma, F1804 1:1000), anti-MYC (Santa Cruz sc 40 1:1000), anti-53BP1 (Novus, NB100-304 1:1000), or anti-RPA2 (Invitrogen PA1-23299 1:500). CY3-(AAACCC)$_3$ PNA probe was used to detect mutant telomere sequence while Alexa488-(AATCCC)$_3$ or CY5-(AATCCC)$_3$ PNA probe was used to detect wild type telomeres. Images acquired using a Zeiss Axio Imager M2 and Axiocam 702 with ZEN 2.6 software. Z-stacks were displayed as maximum intensity projections. For quantification at least 100 nuclei/ condition/experiments were quantified. Figures were assembled with Adobe Illustrator 2024.

### FISH on metaphase spreads
Telomeric FISH on metaphase spreads was performed as previously described[52]. Wild-type telomeric DNA were detected using the following PNA probe: AlexaFluor 488-(AATCCC)3, while mutant telomeric repeats were detected using the following PNA probe: CY3-(AAACCC)3. For quantification at least 250 chromosomes were scored/ condition were quantified. Images were acquired using a Zeiss Axio Imager.

### Western blotting and immunoprecipitation
Cells were lysed in 2× Laemmli buffer and proteins separated by SDS-PAGE on 4–20% TGX Stain-Free gels (Bio-Rad, 4568093). Proteins were transferred to nitrocellulose membranes and probed with antibodies against anti-FLAG (Sigma, F1804 1:1000), MYC (Cell signaling 71D10 1:1000) or tubulin (1:5000; Millipore Sigma, T5168). Detection used HRP- or DyLight-conjugated secondary antibodies and the ChemiDoc™ MP system (Bio-Rad). Immunoprecipitation was performed as previously described[11]. 293 T cells were transfected with the indicated expression plasmids using Lipofectamine 2000 (Life Technologies).

Cells were harvested 24 h post-transfection in NP-40 buffer (50 mM Tris-HCl, pH 7.4; 150 mM NaCl; 5 mM EDTA; 0.5–1% NP-40) supplemented with protease inhibitors (Roche). Supernatants were used for immunoprecipitation using an anti-FLAG antibody (Sigma, F1804 1:100 dilution) and DynaBeads™ A (Invitrogen 10002D). Bound proteins were eluted in Laemmli sample buffer and analyzed by SDS-PAGE.

### In-gel telomere length analysis
Telomere length was analyzed using as previously described[18]. Briefly, mESCs were embedded in agarose plugs and following proteinase K digestion genomic DNA was cut using MboI (NEB), followed by pulsed-field gel electrophoresis on a 1% agarose using a CHEF-DRII system (Bio-Rad). Hybridizations were performed in gel using the following P$^{32}$-ATP end-labeled oligos: [AAACCC]$_4$ to detect mutant telomeric repeats or [AATCCC]$_4$ to detect wild type telomeric repeats under native condition followed by denaturing and re-hybridizations.

### Dot Blot analysis
Genomic DNA was isolated using the Puregene DNA Isolation Kit (Qiagen 1126826) according to the manufacturer's instructions. A total of 25 µg of purified DNA was digested overnight at 37 °C with a cocktail of six restriction enzymes (MspI, Hinfl, RsaI, AluI, MboI, HphI, and MnlI), as previously described[30]. Following digestion, DNA was size-selected using AMPureXP magnetic beads (Beckman Coulter) to remove fragments smaller than 5 kb. Between 25 and 50 ng of the resulting high-molecular-weight DNA was blotted onto a Hybond XL membrane and hybridized overnight at 55 °C with a radiolabeled [AAACCC]$_4$ probe to detect mutant telomeric repeats. Hybridization signals were visualized by phosphorimaging and quantified using ImageQuant software. The blot was then stripped and re-hybridized with a [AATCCC]$_4$ probe to detect wild-type telomeric repeats. Mutant signal intensity was normalized to the wild-type signal to calculate the fold enrichment of mutant repeat incorporation.

### FCR-seq
Genomic DNA was isolated, digested, and telomere sequences were enriched following the same protocol described for the dot blot assay. A total of 100–150 ng of enriched telomeric DNA was then sonicated using a Covaris S220 Ultrasonicator to generate DNA fragments ranging from 300-500 bp. Library preparation for Illumina sequencing was performed using the NEBNext Ultra II kit, following the manufacturer's instructions. Sequencing was carried out using either the P2 100 or P2 220 kit, yielding approximately 40–50 million reads per sample.

Telomeric reads were isolated from the fastq files using the following command:

```
zcat XXX.fastq.gz | grep -E "CTAACCCTAACC|CAAACCCAAA
CC|GTTAGGGTTAGG|GTTTGGGTTGG" > XXX.txt
```

The resulting txt files were analyzed in RStudio to quantify number of reads containing mutants repeats and number of consecutive mutant repeats.

## Statistics and reproducibility

Statistical analyses were conducted using GraphPad Prism version 10.6.1. Unless otherwise indicated, all experiments were analyzed using a two-tailed unpaired *t*-test. The number of independent biological replicates for each experiment is specified in the corresponding figure legends and source data. All representative blots and images shown were derived from at least three independent experiments.

## Reporting summary

Further information on research design is available in the Nature Portfolio Reporting Summary linked to this article.

## Data availability

All sequencing data discussed in this publication have been deposited in NCBI's Gene Expression Omnibus and are accessible through GEO Series accession number GSE304164. Source data are provided with this paper.

## Code availability

The code for the image analysis has been deposited to Github (https://github.com/CCRMicroscopyCore/sandhur) and its accessible via Zenodo using the following link: https://doi.org/10.5281/zenodo.17884033.

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

## Acknowledgements

We thank Travis Stracker and members of the Lazzerini Denchi lab for critical reading of the manuscript and helpful discussion. We thank Michael Kruhlak for assistance with imaging and image quantifications. *Funding*: This work was supported by the following grants from the division of basic science (NCI): 1ZIABC011815 and ZIABC012015. The contributions of the NIH authors are considered Works of the United States Government. The findings and conclusions presented in this paper are those of the authors and do not necessarily reflect the views of the NIH or the U.S. Department of Health and Human Services.

## Author contributions

E.L.D. and R.S. conceived the study, designed the experiments and analyzed the data; R.S., G.T., and S.Y.L. conducted experiments; A.T. developed scripts used to analyze iTAP intensity in interphase cells; E.L.D. and R.S. wrote the paper.

## Funding

## Competing interests

The authors declare no competing interests.
