## [Transparent Peer Review file · Nature Communications]

Control of telomerase recruitment and end protection by independent shelterin components

Corresponding Author: Dr Eros Lazzerini Denchi

Version 0:

Reviewer comments:

Reviewer #1

(Remarks to the Author)

In this study the authors use mouse embryonic stem cells as a model to understand the roles of shelterin proteins POT1 and TPP1 in end protection and end replication. For this purpose, genetic knockout lines for these genes (there are two POT1 proteins POT1A and POT1B in mice) as well as degron-based inducible protein depletion strategies, were used. Using classical techniques and adapting an innovative mutant-template telomerase system to incorporate a new telomeric repeat sequence, this study shows that TPP1 but not POT1 is important for telomerase activity in these cells. Using cell growth and DNA damage response detection at telomeres, the authors show that the POT1 proteins but not TPP1 is important for end protection in these cells. Overall, this is a very easy to follow study with a straightforward rationale and easy to interpret data. However, many conclusions from the first half of the study seem like confirmation of previous reports and those from the second half seem provocative, requiring further affirmations and rationalization.

Major critiques:

1. In the first half of the paper dealing with telomerase activity, all of the major conclusions are consistent with previous literature and there is not much new information. Much of this information can be learned from Abreu et al paper where the authors showed that TPP1 but not POT1 is important for telomerase recruitment in human cells. Additionally, that paper showed TIN2 is important for telomerase recruitment because it recruits TPP1. It also showed that the OB domain of TPP1 is important for telomerase recruitment. As telomerase activity is happening after recruitment, it is not surprising that the current paper sees the phenotypes that they report. Previous studies that used telomere length as a readout of in vivo telomerase activity also came to the same conclusions about the importance of TPP1 in telomerase recruitment. Telomere length measurements of POT1 OB1-deletions have shown that POT1 is an inhibitor of telomerase in human cells. The current study reports a technical advancement of using iTAP setup to read out telomerase activity instead of telomerase recruitment but not many new insights into telomerase activity by shelterin.
2. The use of mutant telomerase combined with FISH and high throughput sequencing to measure telomerase activity in cells is clever. One caveat is that the mutant telomere repeats may affect how TRF1, TRF2 and POT1 bind telomeric DNA. What is known about these proteins binding to TTTGGG or their sequence specificity? If the answer is unknown it seems important to determine the sequence binding specificity in vitro. Is it not surprising that the presence of mutant repeats do not cause TIFs or decrease cell viability? How can this be reasoned in the context of older Blackburn/Stohr lab papers detailing the same and other template mutations?
3. The ability to estimate telomerase processivity in cells is exciting. However It seems possible and quite likely that the TTTGGG repeats are not strongly bound by POT1. If that is true, then it may not be safe to conclude that depleting POT1 protein has no effect on telomerase processivity. If POT1 does not bind efficiently to the newly synthesized TTTGGG repeat, then it doesn't matter if there is POT1 or not in the cells – it would not be able to affect telomerase processivity in either case (which might be what the authors are seeing). Unless there is data in this paper or the literature to show that POT1 binding is not majorly affected by the template mutation, the conclusion of POT1 not affecting telomerase processivity may not be strong.
4. The second half of the paper is interesting with unexpected findings. The take-home message is that although TPP1 is required for robust recruitment of POT1 to telomeres, it is not required for POT1's end protection function. The authors say in the discussion part that their results are consistent with human ES cell studies showing that TPP1 depletion is tolerated but POT1 depletion is not. Looking at the referenced study, it seems like many POT1 mutations in both the TPP1 binding and DNA binding domains are well tolerated. So it seems best to conclude that POT1 function is not essential for the viability of human ES cells. That is not the same as the current paper that argues that POT1 but not TPP1 is important for end

protection. Additionally, the human ES cell paper showed many of the POT1 mutations still show TIFs, which is different from the current paper in which TPP1 depletion doesn't cause significant TIFs.

5. The fact that the importance of TPP1 in end protection is different in MEFs (essential for end protection), whole mice (important for viability), and mouse ES cells (not essential for end protection) is fine if more mechanistic insight into why such differences exist is provided. For example, what if the POT1 expression level dictates its importance – the more limiting POT1 is in a context, the more important it is. This could be tested by looking at expression levels of POT1 in mouse tissues and cell lines. Perhaps the data is already there in databases. Also valuable would be western blots to compare if there is more/less POT1A or POT1B in fibroblasts than stem cells or developing mouse tissues. If not expression, what else do the authors hypothesize to explain the distinct phenotype of stem cells compared to other cells and mice? Whatever their hypothesis, it seems important to test it here.

6. As is, the revised model proposed here is specific for mouse cell lines used. Other in vitro and in vivo studies seem consistent with either model.

Minor critiques:

1. Line 53: correct typographical error (telomeres)

2. Line 57/58: When authors say "both" POT1 and TPP1 stimulate telomerase activity, I assume they mean the TPP1-POT1 complex stimulates telomerase. That is what the cited reference shows. If so, please change the phrasing to make it clear that the complex but not just POT1 stimulates telomerase.

Reviewer #2

(Remarks to the Author)

In their manuscript titled "Control of telomerase recruitment and end protection by independent shelterin components" the authors analyze the contributions of TPP1 and POT1a/b to telomerase recruitment and chromosome end protection in mouse embryonic stem cells. The results demonstrate that TPP1 but not POT1a/b is required for telomerase recruitment using a novel single-cell telomerase activity assay. The authors go on to show that POT1a/b but not TPP1 is required to suppress a telomeric DNA damage response by ATR. This is surprising, because TPP1 is thought to be required for POT1 recruitment to telomeres and thus its end protection function. The authors further support this conclusion by suppressing the DNA damage response caused by loss of POT1a and POT1b by expressing a variant of POT1a that cannot interact with TPP1. The authors clearly demonstrate that knock out of TPP1 does not result in the same DNA damage response as loss of POT1a/b, which supports the notion that some aspect of POT1a/b function does not require its interaction with TPP1. However, the authors do not address whether the specialized functions of POT1a and POT1b could play a role in this observation, and further experiments could address this question to substantially increase the impact of this study.

Major points

Knock out of TPP1 and depletion of POT1a both increase TIF formation to a similar degree, as shown in figure 3C (which needs statistical analysis). While this increase is not as dramatic as loss of both POT1a and POT1b it appears to be at least 5-fold and should not be dismissed and is also ATR dependent (Fig. 3D, also needs statistics). Importantly, knock out of POT1b does not lead to this increase. This suggests that similar levels of ATR signaling occur when TPP1 or POT1a are absent, consistent with the critical role of the POT-hole of POT1a inhibiting the initial activation of ATR by preventing the 9-1-1 complex from associating with the dsDNA-ssDNA junction. Importantly, the authors demonstrate that POT1a re-expression, even without the ability to interact with TPP1, can substantially reduce the ATR response in POT1a/b depleted cells. I am concerned that the TPP1 independence in this experiment is a result of POT1a overexpression, which might allow it to associate with telomeres without the need to bind to TPP1. To address these points the authors should attempt to carry out western blots to compare POT1a expression levels under normal circumstances and in context of the re-expression. While endogenous POT1 expression might be challenging to detect, over-expression might be detectable. In addition, the authors could express POT1a with a POT-hole mutation, with and without TPP1 binding mutation. I believe the authors observations are fully consistent with a model where POT1a requires TPP1 for its function, as evidenced by TPP1 being required for its recruitment and the similar activation of TIF formation the absence of TPP1 and POT1a. In contrast, POT1b might not require TPP1 and directly bind to single-stranded telomeric DNA to prevent RPA binding. When POT1b is absent upstream ATR activation is inhibited by POT1a binding the ssDNA-dsDNA junction and no TIFs are observed. The authors should assess POT1b localization to telomeres in the absence of TPP1.

Minor comments

1. The single-cell telomerase assay might provide more detailed information if the results were presented as a histogram of the number of TTTGGG foci per cell, rather than an arbitrary cutoff of >5 foci per cell.

2. The authors could use the intensity of the FISH signal as a readout of telomerase processivity in addition to the FCR seq approach presented.

Reviewer #3

(Remarks to the Author)

In this manuscript, Sandhu et al. examine the contributions of POT1 and TPP1 to telomerase activity and telomere end protection in mouse embryonic stem cells. The authors apply a clever and elegant approach to measure telomerase activity: transient expression of a mutant TERC followed by detection of the mutant repeat by FISH, dot blot, and sequencing.

They show that TPP1 is required for telomerase activity, while POT1 limits its processivity. Conversely, POT1 depletion results in ATR-dependent telomere deprotection, whereas TPP1 knockout does not. This is surprising, given that POT1 localization depends on TPP1. Consistent with this, the authors find that POT1 fails to form detectable foci in TPP1-deficient cells. Importantly, they demonstrate that POT1's ability to suppress ATR signaling relies on its OB-fold rather than its TPP1-interacting domain, indicating that direct ssDNA binding by POT1 is sufficient for protection.

The results are intriguing and advance our understanding of shelterin function. The study is well designed, and the experiments are convincing.

To further increase the impact, especially for publication in Nature Communications, the authors should extend their mechanistic analysis of how POT1 represses ATR signaling independently of TPP1. For example, are the OB folds of POT1a sufficient for protection, or are additional elements within the POT1C domain required? Similar question for POT1b.

Other comment:

-The experiment in Figure 2D should be performed with POT1a suppression only, as this is likely more quantitative than the sequencing data.

Minor comments:

-The authors use a mutant TERC to measure telomerase activity, leading to the incorporation of mutant telomere repeats, which are likely not efficiently bound by shelterin components, including TPP1 and POT1. That made me wonder if this strategy could bias the analysis, including the processivity measurements. However, I reasoned that the authors likely look at one round of telomerase extension, and thus this approach is not a problem. Nevertheless, the authors should explicitly address this in the results section.

-The order of the panels in supplementary figures S2–S3 is difficult to follow. Consider reorganizing.

-Figure 3D: It appears that all conditions are Pot1a^{-/-}, but this is not correctly indicated in the quantification (red dots).

Version 1:

Reviewer comments:

Reviewer #1

(Remarks to the Author)

The authors have clearly devoted considerable care to addressing the reviewer comments, and their thoughtful engagement is appreciated. The use of the iTAP assay represents a meaningful advance for investigating telomerase activity, offering advantages over traditional approaches focused on recruitment or telomere length. The manuscript is much improved after revision.

Having read the manuscript and seen the new data, I still find the most intriguing aspect to be the suggestion that POT1A can function independently of TPP1 to protect telomeres. This hypothesis stands in contrast to prevailing models in the field and, if confirmed, would represent a significant paradigm shift. To properly substantiate this, I think it could help to clarify some aspects of the current data. The experiments with various POT1A and POT1B constructs are certainly informative, but I am still unclear about the underlying mechanism. Specifically, the observation that a POT1A W-E/F-E mutant unable to recruit TPP1 still rescues TIF formation, while a truncation lacking the entire TPP1 interaction domain does not, suggests an additional function for this region.

Given that in current models, CST complex binding is mediated by POT1B rather than POT1A, it raises the possibility that there may be a novel activity within the TPP1 interaction domain in POT1A. Alternatively, could residual TPP1 protein (for example, a truncation that can still bind POT1A) still be present in cells classified as TPP1 Δ ? It seems important to further characterize what about POT1A in mouse allows it to bypass TPP1 for end protection.

Reviewer #2

(Remarks to the Author)

In this revised manuscript the authors have added a substantial amount of new experiments to assess the role of expression levels and the POT1a "POT-hole" in the observed phenotypes. They have addressed my concerns and I fully support publication of this important manuscript.

Reviewer #3

(Remarks to the Author)

The revised manuscript from Sandhu and colleagues adequately addressed my comments. The authors discussed the potential caveats of the iTAP system. They added a dot-blot quantification of mutTR addition upon POT1a deletion only, showing a minor increase in telomere repeats addition. Most importantly, they showed that POT1a OB folds are not sufficient to provide telomere protection. Altogether, the data provide important insight into the separate functions of POT1 and TPP1 at telomeres, and thus warrant publication into Nature Communications.

Prior to publication:

- There is typo on line 86 (hisher instead of higher)

- Figure 3D was not corrected. The quantification needs to display 4 red dots, as all conditions are POT1adTAG+dTAG

We thank the Reviewers for the thorough and constructive evaluation of our manuscript. We are particularly grateful to Reviewer 1 for recognizing that our study presents a clear rationale, easy-to-interpret data, and for highlighting the technical advancement offered by the iTAP system to measure telomerase activity in cells. We thank Reviewer 2 for noting the strength of our single-cell telomerase assay and for emphasizing the significance of our finding that POT1a/b, but not TPP1, is required to suppress ATR signaling. We also appreciate Reviewer 3's positive assessment of the experimental design, the convincing nature of the data, and the conceptual advance regarding independent functions of POT1 in telomerase regulation and end protection.

The reviewers provided valuable suggestions that have helped us substantially strengthen the mechanistic framework and clarify the interpretation of our results. In the revised manuscript, we have performed additional experiments, expanded several analyses, clarified methodological points, reorganized figures where appropriate, and revised the text for accuracy and precision. All concerns raised during the review have been addressed in detail in our point-by-point responses below.

Reviewer #1:

In this study the authors use mouse embryonic stem cells as a model to understand the roles of shelterin proteins POT1 and TPP1 in end protection and end replication. For this purpose, genetic knockout lines for these genes (there are two POT1 proteins POT1A and POT1B in mice) as well as degron-based inducible protein depletion strategies, were used. Using classical techniques and adapting an innovative mutant-template telomerase system to incorporate a new telomeric repeat sequence, this study shows that TPP1 but not POT1 is important for telomerase activity in these cells. Using cell growth and DNA damage response detection at telomeres, the authors show that the POT1 proteins but not TPP1 is important for end protection in these cells. Overall, this is a very easy to follow study with a straightforward rationale and easy to interpret data. However, many conclusions from the first half of the study seem like confirmation of previous reports and those from the second half seem provocative, requiring further affirmations and rationalization.

We thank the reviewer for their thoughtful summary and positive evaluation of our study's clarity, rationale, and design. We have addressed all major concerns and performed additional experiments to strengthen and further substantiate the unexpected findings presented in the revised manuscript.

1. In the first half of the paper dealing with telomerase activity, all of the major conclusions are consistent with previous literature and there is not much new information. Much of

this information can be learned from Abreu et al paper where the authors showed that TPP1 but not POT1 is important for telomerase recruitment in human cells. Additionally, that paper showed TIN2 is important for telomerase recruitment because it recruits TPP1. It also showed that the OB domain of TPP1 is important for telomerase recruitment. As telomerase activity is happening after recruitment, it is not surprising that the current paper sees the phenotypes that they report. Previous studies that used telomere length as a readout of in vivo telomerase activity also came to the same conclusions about the importance of TPP1 in telomerase recruitment. Telomere length measurements of POT1 OB1-deletions have shown that POT1 is an inhibitor of telomerase in human cells. The current study reports a technical advancement of using iTAP setup to read out telomerase activity instead of telomerase recruitment but not many new insights into telomerase activity by shelterin.

We appreciate the reviewer's point that prior work, including Abreu et al., has demonstrated the importance of TPP1, TIN2, and the TPP1 OB domain in telomerase recruitment, and that POT1 can function as a negative regulator of telomerase activity in human cells. At the same time, several studies have suggested a positive role for POT1 in telomerase processivity in vitro, and recent structural data showing direct POT1–TERT interactions further support this possibility ¹. These findings underscore that the precise role of POT1 in regulating telomerase has remained unresolved, particularly given that TPP1 and POT1 are widely assumed to function as an obligate heterodimer.

We also note that the role of TIN2 in telomerase regulation is complex. Although TIN2 knockdown decreases telomerase recruitment, TIN2 knockdown or haploinsufficiency paradoxically results in telomere elongation ²⁻⁴. Thus, recruitment alone does not fully explain productive telomerase action *in vivo*.

For these reasons, we respectfully disagree that our study simply confirms previous findings. The iTAP assay directly measures telomerase activity in living cells, overcoming the limitations of recruitment-based or exclusively *in vitro* approaches. This strategy allows us to distinguish which shelterin components are required for productive telomerase activity within a physiological chromatin context. In doing so, our work resolves ambiguities that have persisted despite extensive prior studies and provides new insight into shelterin-mediated regulation of telomerase activity.

2. The use of mutant telomerase combined with FISH and high throughput sequencing to measure telomerase activity in cells is clever. One caveat is that the mutant telomere repeats may affect how TRF1, TRF2 and POT1 bind telomeric DNA. What is known about these proteins binding to TTTGGG or their sequence specificity? If the answer is unknown it seems important to determine the sequence binding specificity in vitro. Is it not surprising that the presence of mutant repeats do not cause TIFs or decrease cell

viability? How can this be reasoned in the context of older Blackburn/Stohr lab papers detailing the same and other template mutations?

We thank the reviewer for recognizing the strength of the iTAP assay. The possibility that mutant repeats could interfere with shelterin binding and compromise telomere protection was a central consideration in the design of our system. As the reviewer notes, shelterin components display sequence specificity: TRF1 and TRF2 do not efficiently bind TTTGGG repeats ⁵, and SELEX analyses have shown that the OB1 domain of POT1 has a strict requirement for the canonical TTAGG motif, predicting reduced binding to our mutant repeat ⁶.

To avoid the telomere dysfunction observed in earlier studies using constitutive, high-level expression of mutant TR (including work from the Blackburn laboratory), we incorporated several safeguards. First, we used a tetracycline-inducible mutTR system in mESCs, enabling tightly controlled and low-background expression (Suppl. Fig. 1A). Second, iTAP assays were deliberately limited to 48–72 hours, preventing significant accumulation of mutant repeats that could trigger DNA damage signaling. This approach is consistent with work showing that low-level expression of mutant TR is tolerated without inducing TIFs in human cancer cells ⁷. Moreover, naturally occurring mutations in the TR template region have been reported in vivo ⁸, supporting the notion that limited incorporation of variant repeats is compatible with normal telomere function.

As an additional validation of assay stringency, we deleted endogenous TERC in mutTR-expressing mESCs. Under these conditions, induction of mutTR is highly toxic and leads to rapid growth arrest (Suppl. Fig. 1E), consistent with the telomere dysfunction previously reported upon accumulation of mutant repeats. These observations confirm that mESCs mount a robust response to telomere protection defects and that the limited mutant repeat incorporation under iTAP conditions is indeed well tolerated.

We have clarified these design considerations and supporting data in the revised manuscript. Following text is added to the manuscript text:

“In other cell types constitutive expression of mutTR at higher levels than endogenous hTR has been shown to induce a strong DNA damage response at telomeres. We speculate that, in our system, mutTR expression can be tolerated since it is confined to a short window of time and expressed at relatively low levels compared to endogenous TR. In support of this hypothesis induction of mutTR in cells depleted for endogenous TERC was highly toxic, resulting in growth arrest.”

3. The ability to estimate telomerase processivity in cells is exciting. However It seems possible and quite likely that the TTTGGG repeats are not strongly bound by POT1. If that is true, then it may not be safe to conclude that depleting POT1 protein has no effect

on telomerase processivity. If POT1 does not bind efficiently to the newly synthesized TTTGGG repeat, then it doesn't matter if there is POT1 or not in the cells – it would not be able to affect telomerase processivity in either case (which might be what the authors are seeing). Unless there is data in this paper or the literature to show that POT1 binding is not majorly affected by the template mutation, the conclusion of POT1 not affecting telomerase processivity may not be strong.

We thank the reviewer for highlighting this important point. We agree that POT1 is unlikely to efficiently bind the newly synthesized TTTGGG repeats and therefore would not be expected to directly modulate telomerase processivity on this mutant sequence. Because of this, we do not interpret our data as evidence that POT1 cannot influence processivity per se, but rather as evidence that the increased mutant repeat incorporation observed in POT1-depleted cells reflects increased telomerase engagement.

Our analysis specifically compares telomerase activity in the presence or absence of POT1 prior to the first round of mutant telomerase extension. As shown in Figure 3D and Supplementary Figure 3B, POT1 depletion consistently results in a substantial increase in the number of mutant repeats added. Since POT1 is the only altered factor in an otherwise wild-type telomeric context before mutant extension begins, this increase must arise from (i) elevated telomerase access to telomeres, or (ii) altered processivity. The nucleotide-resolution FCR-seq data strongly support the former: we observe more frequent initial engagement events in POT1-depleted cells, while the amount of repeat addition per engagement is unchanged.

Thus, our findings indicate that POT1 primarily restricts telomerase access to telomeres. We fully acknowledge, as the reviewer astutely notes, that because POT1 does not bind efficiently to the mutant sequence, our assay likely underestimates any role POT1 might play in regulating processivity on canonical repeats. We have clarified this limitation explicitly in the revised manuscript by adding following text:

“POT1 cannot bind efficiently to the mutant telomeric sequences, suggesting that our assay may underestimate the effect of POT1 on telomerase once this enzyme is engaged with the substrate. Nevertheless, in our inducible experimental setting we are primarily assessing the first round of incorporation of mutant repeats within a wild-type telomeric sequence, thus limiting as much as possible this caveat”.

4. The second half of the paper is interesting with unexpected findings. The take-home message is that although TPP1 is required for robust recruitment of POT1 to telomeres, it is not required for POT1's end protection function. The authors say in the discussion part that their results are consistent with human ES cell studies showing that TPP1 depletion is tolerated but POT1 depletion is not. Looking at the referenced study, it seems like many POT1 mutations in both the TPP1 binding and DNA binding domains are well tolerated. So it seems best to conclude that POT1 function is not essential for the viability of human ES cells. That is not the same as the current paper that argues that POT1 but

not TPP1 is important for end protection. Additionally, the human ES cell paper showed many of the POT1 mutations still show TIFs, which is different from the current paper in which TPP1 depletion doesn't cause significant TIFs.

We thank the reviewer for this thoughtful comment and for drawing attention to the human ES cell data. We apologize for the incorrect citation in our original text; this has been corrected in the revised manuscript ⁹.

With respect to the essentiality of POT1 in human ES cells, we believe that the published data remain fully consistent with our interpretation. In human ES cells, complete loss-of-function alleles of POT1, such as frameshifts or truncations predicted to eliminate both DNA binding and TPP1 interaction, are not recovered, indicating that total POT1 deficiency is not tolerated. This aligns with our findings in mESCs, where simultaneous depletion of POT1a and POT1b leads to rapid telomere deprotection and loss of viability.

We acknowledge the reviewer's correct observation that several missense mutations within the POT1 OB folds or TPP1-interaction domain are compatible with survival in human ES cells. However, it's likely that these variants have reduced, but not abolished, DNA-binding activity and thus are best interpreted as hypomorphic alleles. Indeed, mutations such as F62A, which fully abrogate POT1's ability to bind single-stranded telomeric DNA, are not tolerated in human ES cells. This pattern suggests that ES cells require a minimal threshold of POT1-mediated end protection, rather than the complete dispensability of POT1 function.

Regarding the difference in TIF formation between POT1 and TPP1 perturbations: the human ES cell study reports that many POT1 hypomorphic alleles still induce detectable TIFs, whereas we do not observe significant TIF accumulation upon TPP1 loss in mESCs. We interpret this discrepancy as reflecting the retained (though partial) ssDNA-binding activity of the hypomorphic human POT1 variants. In contrast, TPP1 loss in mESCs reduces POT1 recruitment but does not eliminate POT1's intrinsic ability to bind ssDNA, allowing sufficient residual end protection.

Taken together, these data support our conclusion that TPP1 is dispensable for POT1-mediated end protection, whereas POT1 function itself is essential for telomere integrity and viability.

5. The fact that the importance of TPP1 in end protection is different in MEFs (essential for end protection), whole mice (important for viability), and mouse ES cells (not essential for end protection) is fine if more mechanistic insight into why such differences exist is provided. For example, what if the POT1 expression level dictates its importance – the more limiting POT1 is in a context, the more important it is. This could be tested by looking at expression levels of POT1 in mouse tissues and cell lines. Perhaps the data is already

there in databases. Also, valuable would-be western blots to compare if there is more/less POT1A or POT1B in fibroblasts than stem cells or developing mouse tissues. If not expression, what else do the authors hypothesize to explain the distinct phenotype of stem cells compared to other cells and mice? Whatever their hypothesis, it seems important to test it here.

We thank the reviewer for this thoughtful comment. The question of why telomere end protection differs between pluripotent and differentiated cells is an exciting one.

Because the role of TPP1 in telomere protection depends primarily on its ability to recruit POT1 to telomeres, one possible explanation is that higher levels of POT1 in ES cells might bypass the requirement for TPP1. Indeed, transcriptomic data indicate variable expression of both POT1a and POT1b across cell types, with POT1a expressed at higher levels during early development compared to several differentiated cell types. To test whether this translates into a significant difference in protein levels, we differentiated mESCs carrying an endogenously tagged POT1a allele into neural progenitor cells (NPCs). We chose NPCs because transcriptional POT1a levels are low in these cells compared to mESCs (see figure below). In the resulting isogenic cell lines, we performed western blot analysis to compare POT1a protein levels between comparable numbers of undifferentiated and differentiated cells. As shown in the accompanying data, we did not detect a marked difference in POT1a protein abundance between these stages, suggesting that altered POT1 levels are unlikely to be the primary driver of the distinct telomere protection phenotype in ES cells. Nevertheless, a negative result in this type of experiment cannot be interpreted as conclusive, and additional experiments would be needed to rigorously exclude a contribution of POT1a/b levels to the TPP1 requirement in end protection. We think this falls beyond the scope of the present manuscript, and we would like to share our data with the scientific community as soon as possible so that others, if interested, might also address this challenging question.

Another hypothesis we are exploring is that a critical difference between mESCs and differentiated cell types is the level of available RPA complex to detect single-stranded telomeric DNA. mESCs display unique cell-cycle characteristics, spending most of their time in S phase and exhibiting elevated levels of single-stranded DNA throughout the genome¹⁰⁻¹². We speculate that in this context, a substantial fraction of RPA is sequestered at non-telomeric sites, thereby reducing its competition with POT1 at telomeres and rendering TPP1-mediated recruitment less critical. To test this hypothesis, we attempted to overexpress RPA in mESCs, similar to what was done by other groups¹³. These experiments yielded a negative result; however, the level of ectopic RPA in mESCs was extremely low, rendering the interpretation of these experiments

inconclusive. As above, a negative result in this type of experiment cannot be considered definitive, and additional work will be required to determine whether RPA availability and competition with POT1 underlie the distinct TPP1 requirement in pluripotent versus differentiated cells. We believe that a comprehensive dissection of this question is beyond the scope of the current study, but we hope that by presenting our data we will stimulate further investigation by others in the field.

6. As is, the revised model proposed here is specific for mouse cell lines used. Other *in vitro* and *in vivo* studies seem consistent with either model.

We thank the reviewer for this comment. We agree that our current model is derived from experiments performed in mouse ES cells and that additional work will be required to determine the extent to which this framework applies to other cell types and organisms. Importantly, our data do not exclude the possibility that alternative modes of POT1 loading or end protection operate in differentiated cells or *in vivo*.

At the same time, we note that several published studies both *in vitro* and *in vivo* are compatible with aspects of our model, including the observation that POT1 can bind single-stranded telomeric DNA independently of TPP1 and the notion that the relative contributions of TPP1 and POT1 to end protection may vary depending on cellular context. Our work provides a mechanistic explanation for how POT1 can retain its essential end protection function even when TPP1-mediated recruitment is compromised, at least in pluripotent cells.

Minor critiques:

1. Line 53: correct typographical error (telomeres)

The issue has been fixed.

2. Line 57/58: When authors say “both” POT1 and TPP1 stimulate telomerase activity, I assume they mean the TPP1-POT1 complex stimulates telomerase. That is what the cited reference shows. If so, please change the phrasing to make it clear that the complex but not just POT1 stimulates telomerase.

The issue has been fixed.

Reviewer #2:

In their manuscript titled “Control of telomerase recruitment and end protection by independent shelterin components” the authors analyze the contributions of TPP1 and POT1a/b to telomerase recruitment and chromosome end protection in mouse embryonic stem cells. The results demonstrate that TPP1 but not POT1a/b is required for telomerase recruitment using a novel single-cell telomerase activity assay. The authors go on to show that POT1a/b but not TPP1 is required to suppress a telomeric DNA damage response

by ATR. This is surprising, because TPP1 is thought to be required for POT1 recruitment to telomeres and thus its end protection function. The authors further support this conclusion by suppressing the DNA damage response caused by loss of POT1a and POT1b by expressing a variant of POT1a that cannot interact with TPP1. The authors clearly demonstrate that knock out of TPP1 does not result in the same DNA damage response as loss of POT1a/b, which supports the notion that some aspect of POT1a/b function does not require its interaction with TPP1. However, the authors do not address whether the specialized functions of POT1a and POT1b could play a role in this observation, and further experiments could address this question to substantially increase the impact of this study.

We thank the reviewer for their encouraging assessment of our manuscript and for the thoughtful and constructive suggestions. We are pleased that the reviewer recognizes the strength of our findings showing that the TPP1–POT1 interaction is dispensable for both telomerase function and telomere protection in mESCs.

The reviewer raises the importance of teasing the the specialized functions of POT1a and POT1b.

To address this point, we have performed additional experiments to delineate the distinct contributions of POT1a and POT1b to telomere end protection. Our new data show that although both POT1a and POT1b are able to rescue the growth defect of Pot1a/b-deficient cells, they differ markedly in their ability to suppress ATR-dependent telomere dysfunction (Fig 4E & 4F). Specifically, POT1a fully suppresses TIF formation to wild-type levels, whereas POT1b provides only partial rescue. Importantly, we demonstrate that this functional distinction arises from the “POT-hole” residues unique to POT1a. Substituting these residues in POT1a with the corresponding POT1b residues abolishes its end-protection activity and causes POT1a to behave like POT1b (Fig. 4C, 4E & 4F, Suppl Fig. 5A & B).

These findings provide mechanistic insight into the specialized roles of POT1a and POT1b and clarify why TPP1 loss does not phenocopy POT1 deficiency in mESCs. We have incorporated these new experiments and their interpretation into the revised manuscript.

Major points

Knock out of TPP1 and depletion of POT1a both increase TIF formation to a similar degree, as shown in figure 3C (which needs statistical analysis). While this increase is not as dramatic as loss of both POT1a and POT1b it appears to be at least 5-fold and should not be dismissed and is also ATR dependent (Fig. 3D, also needs statistics). Importantly, knock out of POT1b does not lead to this increase. This suggests that similar levels of ATR signaling occur when TPP1 or POT1a are absent, consistent with the critical role of the POT-hole of POT1a inhibiting the initial activation of ATR by preventing the 9-1-1 complex from associating with the dsDNA-ssDNA junction. Importantly, the authors demonstrate that POT1a re-expression, even without the ability to interact with TPP1, can substantially reduce the ATR response in POT1a/b depleted cells. I am concerned that

the TPP1 independence in this experiment is a result of POT1a overexpression, which might allow it to associate with telomeres without the need to bind to TPP1. To address these points the authors should attempt to carry out western blots to compare POT1a expression levels under normal circumstances and in context of the re-expression. While endogenous POT1 expression might be challenging to detect, over-expression might be detectable. In addition, the authors could express POT1a with a POT-hole mutation, with and without TPP1 binding mutation. I believe the authors observations are fully consistent with a model where POT1a requires TPP1 for its function, as evidenced by TPP1 being required for its recruitment and the similar activation of TIF formation the absence of TPP1 and POT1a. In contrast, POT1b might not require TPP1 and directly bind to single-stranded telomeric DNA to prevent RPA binding. When POT1b is absent upstream ATR activation is inhibited by POT1a binding the ssDNA-dsDNA junction and no TIFs are observed. The authors should assess POT1b localization to telomeres in the absence of TPP1.

We thank the reviewer for this detailed and insightful comment, which helped us clarify several aspects of our analysis. We address each component of the comment separately below.

1. TPP1 knockout and TIF formation

We would first like to clarify a source of confusion: TPP1 knockout cells do not exhibit increased TIF levels. The apparent increase noted by the reviewer resulted from a labeling error in Figure 3C, where TPP1^{-/-}POT1a^{dTAG} cells were inadvertently labeled as TPP1^{-/-}. We sincerely apologize for this oversight. The corrected figure clearly shows that TPP1 loss alone does not induce telomere deprotection, fully consistent with our conclusion that TPP1 is dispensable for end protection in mESCs.

2. Distinct roles of POT1a and POT1b in ATR suppression

After correction of Figure 3C, our results indicate:

Loss of POT1a, with or without TPP1, leads to a robust increase in TIFs.

Loss of POT1b does not cause any detectable telomere deprotection.

Statistical analysis of these data (now added to Fig. 3C and 3D) confirms these conclusions.

We interpret these findings in light of the proposed role of POT1a in binding the telomeric ds-ss junction via the “POT-hole” residues¹⁴. Several lines of evidence from our study support this model:

Both POT1a and POT1b require TPP1 for efficient accumulation at telomeres (Fig. 4A-B, Suppl. Fig. 4C). Thus, the functional differences cannot be attributed to differential TPP1-binding or recruitment.

POT1a depletion induces TIFs irrespective of TPP1 status, demonstrating that POT1a's end-protection activity is intrinsic and does not rely on TPP1-mediated telomeric loading (Fig. 3C).

POT1a, but not POT1b, fully suppresses TIFs when re-expressed in POT1a/b-deficient cells, and crucially, POT1a alleles lacking the POT-hole residues fail to suppress TIFs and behave like POT1b (Fig. 4E).

These new experiments directly address the reviewer's suggestion to compare POT1a variants with and without TPP1-binding capacity and with POT-hole mutations. The relevant data have been added to the revised manuscript (Fig. 3C, 4C-F, Suppl Fig. 4A-C).

3. Concern that POT1a overexpression might bypass the need for TPP1

We appreciate this important concern, and we have addressed it as follows:

Although endogenous POT1a is difficult to detect due to the lack of reliable antibodies, we can robustly detect all ectopically expressed POT1a variants via their N-terminal epitope tags. These data (Suppl. Fig. 5A) confirm comparable levels of expression across all mutants, including the TPP1-binding-deficient alleles. Most critically, our conclusions are reinforced by the strikingly different phenotypes of TPP1-null versus POT1-null cells. TPP1-null mESCs maintain end protection, whereas POT1a/b-null mESCs exhibit severe ATR activation and rapid loss of viability, demonstrating that the essential end-protection activity resides within POT1a/b themselves.

Together, these findings strongly argue that the observed TPP1 independence is not an artifact of overexpression but instead reflects the intrinsic ability of POT1a to engage telomeric ssDNA and to block ATR activation, even when TPP1-mediated recruitment is absent.

4. POT1b localization in the absence of TPP1

In response to the reviewer's suggestion, we have now assessed POT1b localization in TPP1-null (Suppl. Fig. 4C). Similar to POT1a, POT1b fails to accumulate efficiently at telomeres in the absence of TPP1. This demonstrates that both POT1a and POT1b require TPP1 for robust telomeric recruitment, despite their distinct functional roles in end protection.

Minor comments

1. The single-cell telomerase assay might provide more detailed information if the results were presented as a histogram of the number of TTTGGG foci per cell, rather than an arbitrary cutoff of >5 foci per cell.

See below.

2. The authors could use the intensity of the FISH signal as a readout of telomerase processivity in addition to the FCR seq approach presented

We agree that more quantitative visualization of single-cell telomerase activity would enhance the interpretability of the iTAP assay. To address the reviewer's comment, we performed automated quantification of both the number and average intensity of mutant telomeric foci per cell. As shown in Supplementary Figure 1D-E, mutTR induction for 48–62 hours leads to a significant increase in both the number and intensity of mutant foci, validating that fluorescence-based iTAP measurements capture robust changes in telomerase activity.

However, while the assay readily detects major differences, it is currently less sensitive in distinguishing more subtle changes compared with the FCR-seq approach. For example, although POT1a depletion results in a clear and statistically significant increase in the number of mutant foci per cell, the corresponding change in foci intensity does not reach statistical significance. Because the precise quantitative relationship between FISH-based foci intensity and telomerase processivity has not yet been fully defined, we prefer to avoid over-interpretation of these fluorescence-based metrics.

For these reasons, we consider FCR-seq to remain the more sensitive and quantitative method for assessing subtle differences in telomerase activity and processivity in the current study. We have clarified this point in the revised manuscript.

Reviewer #3:

In this manuscript, Sandhu et al. examine the contributions of POT1 and TPP1 to telomerase activity and telomere end protection in mouse embryonic stem cells. The authors apply a clever and elegant approach to measure telomerase activity: transient expression of a mutant TERC followed by detection of the mutant repeat by FISH, dot

blot, and sequencing. They show that TPP1 is required for telomerase activity, while POT1 limits its processivity. Conversely, POT1 depletion results in ATR-dependent telomere deprotection, whereas TPP1 knockout does not. This is surprising, given that POT1 localization depends on TPP1. Consistent with this, the authors find that POT1 fails to form detectable foci in TPP1-deficient cells. Importantly, they demonstrate that POT1's ability to suppress ATR signaling relies on its OB-fold rather than its TPP1-interacting domain, indicating that direct ssDNA binding by POT1 is sufficient for protection. The results are intriguing and advance our understanding of shelterin function. The study is well designed, and the experiments are convincing. To further increase the impact, especially for publication in Nature Communications, the authors should extend their mechanistic analysis of how POT1 represses ATR signaling independently of TPP1. For example, are the OB folds of POT1a sufficient for protection, or are additional elements within the POT1C domain required? Similar question for POT1b.

We thank the reviewer for their positive assessment of our study and for recognizing the value of our iTAP assay for quantifying telomerase activity.

We appreciate the suggestion to further investigate the mechanism by which POT1 represses ATR signaling independently of TPP1. To address this point, we performed additional experiments to test whether the OB-fold domain of POT1a alone is sufficient for telomere end protection. As shown in Fig. 4E and Supplementary Fig. 5A, B, expression of the POT1a OB-fold domains failed to rescue TIF formation upon POT1a/b depletion, indicating that ssDNA binding alone is not sufficient for end protection. These results suggest that additional regions beyond the OB-fold are required, although it remains unclear whether these regions contribute to proper folding or stability of the OB domain, or mediate other essential interactions independent of TPP1. We have also included new characterization of POT1b's role in suppressing TIFs, showing that POT1b is unable to fully prevent TIF formation, likely due to its inability to bind the ds-ss telomeric junction.

The experiment in Figure 2D should be performed with POT1a suppression only, as this is likely more quantitative than the sequencing data.

We thank reviewer for the suggest, in our revised manuscript we have added this data (Suppl Fig 4B).

Minor comments:

-The authors use a mutant TERC to measure telomerase activity, leading to the incorporation of mutant telomere repeats, which are likely not efficiently bound by shelterin components, including TPP1 and POT1. That made me wonder if this strategy could bias the analysis, including the processivity measurements. However, I reasoned that the authors likely look at one round of telomerase extension, and thus this approach is not a problem. Nevertheless, the authors should explicitly address this the results section.

We thank the reviewer for these valuable comments. We agree that incorporation a mutant repeat will lead to a reduction in affinity for shelterin components, including TPP1 and POT1. Importantly, as the reviewer correctly points out, the iTAP assay is designed to capture the first round of telomerase engagement with wild-type telomeres. Under our experimental conditions, mutant repeat incorporation remains limited, which minimizes any potential bias arising from altered shelterin binding to extended tracts of mutant sequence.

With respect to processivity measurements, we fully acknowledge that reduced POT1 binding to mutant repeats could limit our ability to detect contributions of POT1 to extension length. We have now explicitly stated this caveat in the revised results section and clarified that iTAP predominantly reports on recruitment and initial engagement rather than on downstream rounds of repeat addition. Following text is added to the result section:

“POT1 cannot bind efficiently to the mutant telomeric sequences, suggesting that our assay may underestimate the effect of POT1 on telomerase once this enzyme is engaged with the substrate. Nevertheless, in our inducible experimental setting we are primarily assessing the first round of incorporation of mutant repeats within a wild-type telomeric sequence, thus limiting as much as possible this caveat”.

-The order of the panels in supplementary figures S2–S3 is difficult to follow. Consider reorganizing.

We thank the reviewer for this comment. These figures were probably too dense with data, we removed the chromatograms to simplify the layout.

-Figure 3D: It appears that all conditions are Pot1a^{-/-}, but this is not correctly indicated in the quantification (red dots).

We thank reviewer for pointing this out. This mistake has been corrected in revised manuscript.

References:

1. Sekne, Z., Ghanim, G.E., van Roon, A.M. & Nguyen, T.H.D. Structural basis of human telomerase recruitment by TPP1-POT1. *Science* **375**, 1173–1176 (2022).
2. Abreu, E. *et al.* TIN2-Tethered TPP1 Recruits Human Telomerase to Telomeres. *Mol Cell Biol* **30**, 2971–2982 (2010).
3. Ye, J.Z.S. & de Lange, T. TIN2 is a tankyrase 1 PARP modulator in the TRF1 telomere length control complex. *Nat Genet* **36**, 618–623 (2004).
4. Schmutz, I. *et al.* TIN2 is a haploinsufficient tumor suppressor that limits telomere length. *Elife* **9** (2020).

5. Tham, C.Y. *et al.* High-throughput telomere length measurement at nucleotide resolution using the PacBio high fidelity sequencing platform. *Nat Commun* **14**, 281 (2023).
6. Choi, K.H. *et al.* The OB-fold domain 1 of human POT1 recognizes both telomeric and non-telomeric DNA motifs. *Biochimie* **115**, 17–27 (2015).
7. Frank, A.K. *et al.* The Shelterin TIN2 Subunit Mediates Recruitment of Telomerase to Telomeres. *Plos Genet* **11** (2015).
8. Hinchie, A.M. *et al.* A persistent variant telomere sequence in a human pedigree. *Nature Communications* **15** (2024).
9. Boyle, J.M. *et al.* Telomere length set point regulation in human pluripotent stem cells critically depends on the shelterin protein TPP1. *Mol Biol Cell* **31**, 2583–2596 (2020).
10. Liu, L., Michowski, W., Kolodziejczyk, A. & Sicinski, P. The cell cycle in stem cell proliferation, pluripotency and differentiation. *Nat Cell Biol* **21**, 1060–1067 (2019).
11. Ahuja, A.K. *et al.* A short G1 phase imposes constitutive replication stress and fork remodelling in mouse embryonic stem cells. *Nat Commun* **7**, 10660 (2016).
12. Banath, J.P. *et al.* Explanation for excessive DNA single-strand breaks and endogenous repair foci in pluripotent mouse embryonic stem cells. *Exp Cell Res* **315**, 1505–1520 (2009).
13. Ortega, P. *et al.* Mechanism of DNA replication fork breakage and PARP1 hyperactivation during replication catastrophe. *Sci Adv* **11**, eadu0437 (2025).
14. Tesmer, V.M., Brenner, K.A. & Nandakumar, J. Human POT1 protects the telomeric ds-ss DNA junction by capping the 5' end of the chromosome. *Science* **381**, 771–778 (2023).

Response to Reviewers

We thank the reviewers for their careful evaluation of the revised manuscript and for their constructive and supportive comments. We are pleased that all reviewers support publication. Below we address the remaining points raised prior to final acceptance.

Reviewer #1

Reviewer comment:

“The authors have clearly devoted considerable care to addressing the reviewer comments, and their thoughtful engagement is appreciated. The use of the iTAP assay represents a meaningful advance for investigating telomerase activity, offering advantages over traditional approaches focused on recruitment or telomere length. The manuscript is much improved after revision.

Having read the manuscript and seen the new data, I still find the most intriguing aspect to be the suggestion that POT1A can function independently of TPP1 to protect telomeres. This hypothesis stands in contrast to prevailing models in the field and, if confirmed, would represent a significant paradigm shift. To properly substantiate this, I think it could help to clarify some aspects of the current data. The experiments with various POT1A and POT1B constructs are certainly informative, but I am still unclear about the underlying mechanism. Specifically, the observation that a POT1A W-E/F-E mutant unable to recruit TPP1 still rescues TIF formation, while a truncation lacking the entire TPP1 interaction domain does not, suggests an additional function for this region.

Given that in current models, CST complex binding is mediated by POT1B rather than POT1A, it raises the possibility that there may be a novel activity within the TPP1 interaction domain in POT1A. Alternatively, could residual TPP1 protein (for example, a truncation that can still bind POT1A) still be present in cells classified as TPP1 Δ ? It seems important to further characterize what about POT1A in mouse allows it to bypass TPP1 for end protection.”

We thank the reviewer for their thoughtful and positive assessment of the revised manuscript and for highlighting the conceptual significance of our findings.

The reviewer notes that the ability of POT1A to protect telomeres in the absence of TPP1 challenges prevailing models and raises important mechanistic questions. We agree that this is a particularly intriguing aspect of our study. As pointed out, the differential behavior of the POT1A W-E/F-E mutant compared with a mutant completely lacking the TPP1-interaction domain suggests that this region of POT1A may contribute to telomere end protection beyond simply mediating TPP1 binding. We have intentionally limited our conclusions to what is directly supported by the data. Accordingly, in the revised manuscript we frame POT1A's TPP1-independent function as evidence for a previously unappreciated protective activity, while

acknowledging that the precise molecular mechanism remains to be defined. Further dissection of these mechanisms will be the focus of future work.

Reviewer #2

Reviewer comment:

“In this revised manuscript the authors have added a substantial amount of new experiments to assess the role of expression levels and the POT1a ‘POT-hole’ in the observed phenotypes. They have addressed my concerns and I fully support publication of this important manuscript.”

We thank the reviewer for their positive evaluation of the revised manuscript and for their support of publication. We are pleased that the additional experiments addressing POT1A expression levels and the “POT-hole” resolved the reviewer’s concerns.

Reviewer #3

Reviewer comment:

“The revised manuscript from Sandhu and colleagues adequately addressed my comments. The authors discussed the potential caveats of the iTAP system. They added a dot-blot quantification of mutTR addition upon POT1a deletion only, showing a minor increase in telomere repeats addition. Most importantly, they showed that POT1a OB folds are not sufficient to provide telomere protection. Altogether, the data provide important insight into the separate functions of POT1 and TPP1 at telomeres, and thus warrant publication into Nature Communications.

Prior to publication:

– There is typo on line 86 (hisher instead of higher)

– Figure 3D was not corrected. The quantification needs to display 4 red dots, as all conditions are POT1adTAG+dTAG”

We thank the reviewer for their supportive comments and for recognizing the added experiments and clarifications.

We have addressed the remaining minor points as follows:

Typographical error: The typo on line 86 (“hisher”) has been corrected to “higher.” Figure 3D: The quantification has been corrected to display four red data points, consistent with all conditions being POT1a^{dTAG} + dTAG.

We believe these final adjustments fully address the remaining comments, and we thank the reviewers and editors for their careful and constructive guidance.